# GUARD: Guided Unlearning and Retention via Data Attribution for Large Language Models

## Abstract

Unlearning in large language models (LLMs) is becoming increasingly important due to regulatory compliance, copyright protection, and privacy concerns. However, a key challenge in LLM unlearning is *unintended forgetting*, where the removal of specific data inadvertently impairs the utility of the model and its retention of valuable, desired information. While prior work has primarily focused on architectural innovations, the influence of data-level factors on unlearning performance remains underexplored. As a result, existing methods often suffer from degraded retention when forgetting high-impact data. To address this problem, we propose GUARD — a novel framework for Guided Unlearning And Retention via Data attribution. At its core, GUARD introduces a lightweight proxy data attribution metric tailored for LLM unlearning, which quantifies the "alignment" between the Forget and Retain sets while remaining computationally efficient. Building on this, we design a novel unlearning objective that assigns *adaptive, nonuniform unlearning weights to samples,* inversely proportional to their proxy attribution scores. Through such a reallocation of unlearning power, GUARD mitigates unintended retention loss. We also provide rigorous theoretical guarantees that GUARD significantly improves retention while maintaining forgetting metrics comparable to prior methods. Extensive experiments on the TOFU and MUSE benchmarks across multiple LLM architectures demonstrate that GUARD reduces utility sacrifice on the TOFU Retain Set by up to 194.92% in terms of Truth Ratio when forgetting 10% of the training data, and improves knowledge retention on the MUSE NEWS Retain Set by 16.20%, with comparable or very moderate increases in privacy loss compared to state-of-the-art methods.

## 1 Introduction

Although LLMs have demonstrated remarkable domain transferability and achieved state-of-the-art performance across a wide range of natural language processing tasks Ouyang et al. (2022); Wei et al. (2022); Touvron et al. (2023); Wu et al. (2023); Liang et al. (2022), their deployment has raised serious privacy and security concerns. LLMs are typically trained on massive corpora of data that may contain sensitive, private, or proprietary information, which increases the risk of unintended data leakage and misuse Li et al. (2021); Shi et al. (2023); Li et al. (2024b); Yang et al. (2023) (for example, LLMs were shown to memorize and regenerate specific pieces of sensitive content, such as Social Security Numbers (SSNs) Carlini et al. (2021); Huang et al. (2022)). These challenges have prompted a growing interest in developing mechanisms for the selective removal — or *unlearning* — of specific information from LLMs.

Machine Unlearning in LLMs provides a practical alternative to avoid model retraining when removing specific information, in compliance with privacy and regulatory requirements Yao et al. (2024b); Kumar et al. (2022); Chen & Yang (2023); Maini et al. (2024a); Liu & et al. (2024). Despite many advances in the domains of computational efficiency and key forget parameter identification Yao & et al. (2023); Maini & et al. (2024); Zhang & et al. (2024a); Li & et al. (2024); Shi & et al. (2024); Choi & et al. (2024); Wang et al. (2024a); Jia et al. (2024); Niu et al. (2025); Wei et al. (2025), fine-tuning-based LLM unlearning methods frequently suffer from *excessive forgetting* and *retention degradation* Zhang & et al. (2024b), compromising the utility of the retained model knowledge. Similarly, inference-time LLM unlearning techniques Pawelczyk & et al. (2023); Huang & et al.

(2024); Thaker & et al. (2024); Liu et al. (2024); Ji et al. (2024), which avoid model updates and are hence computationally attractive, do not guarantee actual forgetting, as sensitive information remains embedded within the unchanged model parameters (for related works, see Appendix B.1).

More broadly, the current limitations of LLM unlearning methods include one or more of the following: *a) degraded retention*, which refers to the fact that unlearning can severely degrade performance on the retained data Zhang & et al. (2024b;a); Li & et al. (2024); *b) computational inefficiency,* which arises when requiring auxiliary LLMs for guidance or evaluation Ji et al. (2024); *c) privacy leakage*, which results in failure to guarantee acceptable removal of sensitive information Liu et al. (2024); Pawelczyk & et al. (2023); Huang & et al. (2024); and, *d) lack of theoretical performance guarantees.* To address the above problems, we propose a novel unlearning method that ensures *increased utility, effective forgetting, improved computational efficiency, and formal theoretical guarantees for both retention and forgetting.*

**Increased utility and effective forgetting.** Existing unlearning methods use unprincipled allocation of model unlearning weights across forget samples. For example, prior fine-tuning-based approaches typically construct the unlearning objective by averaging per-sample losses with retention-unrelated weights that are often uniform or determined solely by the forget sample itself, without considering the influence of the sample on model performance over the retained set. Neglecting the attributional impact of a sample leads to unintentional forgetting of useful knowledge and consequent utility loss: For example, unlearning a training instance from a particular class can disproportionately degrade performance on that same class when compared to unlearning unrelated instances Koh & Liang (2017). Our solution involves a novel data-attribution-driven unlearning framework termed *GUARD* (**G**uided **U**nlearning **A**nd **R**etention via **D**ata attribution), which explicitly incorporates data attribution metrics into the unlearning objective.

GUARD dynamically adjusts unlearning weights based on the influence of each forget sample on the performance of the model over the retained data. More precisely, GUARD allocates unlearning weights in such a way that the data-wise loss weight is inversely proportional to the sample-wise contribution to retention, reducing the likelihood of unintentionally degrading of retained knowledge. To ensure effective forgetting, we also introduce a temperature-controlled reverse unification mechanism that regulates the variance of unlearning weights across samples. This principled reallocation strategy enables GUARD to both mitigate retention degradation by down-weighting high-retention-impact samples, and to maintain effective forgetting by carefully calibrating the degree of adjustment.

**Improved computational efficiency.** Since GUARD has to perform efficient data attribution on large LLM models, one has to redesign existing methods. Conventional data attribution methods estimate the influence of individual training samples on overall model performance by comparing model outputs with and without the inclusion of each sample Koh & Liang (2017); Ellis et al. (2022); Ingram et al. (2023). However, these approaches typically require model retraining or Hessian-based approximations, making them computationally prohibitive for LLMs. Moreover, evaluating the influence of samples based on global performance contradicts the unlearning objective which aims to *decrease* the performance on the Forget Set and simultaneously *preserve* or *improve* it on the Retain Set, rather than optimize performance across the whole dataset (see Appendix B.2). *Our solution to this problem is a novel attribution scores tailored to LLM unlearning.* Inspired by gradient-based sample-wise representations Nguyen et al. (2025); Wang et al. (2025b), we define the *retention attribution score* of a forget sample as the inner product between its gradient and the gradient averaged over all retain samples. This measure captures the alignment between the knowledge encoded by a forget sample and that of the retain distribution. It also avoids the computational burden of retraining or Hessian estimation inherent to prior attribution techniques. Furthermore, by explicitly separating forget and retain knowledge before computing inter-sample influence, the proposed score effectively quantifies the contribution of a sample to retention utility and facilitates retention-aware unlearning.

**Theoretical Guarantees**. We provide a rigorous theoretical analysis demonstrating that, compared to conventional unlearning baselines such as Gradient Ascent Yao & et al. (2023), GUARD offers the following key benefits: (1) it effectively reduces the loss on the Retain Set, leading to enhanced *absolute retention*; (2) it maintains comparable degradation on the Forget Set, ensuring strong *absolute forgetting*; and (3) it significantly reduces the retention-performance tradeoff, improving *relative retention efficiency*, i.e., retention preserved per unit of forgetting incurred.

**Empirical Validation.** To empirically validate our claims, we conduct extensive experiments on the TOFU Maini et al. (2024b) and MUSE Shi et al. (2024a) benchmarks, two standardized evaluation suites designed specifically for LLM unlearning. Across a range of LLM architectures and evaluation protocols, GUARD consistently and substantially outperforms existing methods both with respect to absolute and relative retention while it maintains comparable levels of forgetting. Notably, GUARD reduces utility degradation on TOFU's Retain Set by up to 194.92% in terms of the Truth Ratio when forgetting 10% of the training data. Also, on the MUSE NEWS Retain Set, GUARD enhances knowledge retention by 16.20%.

**Summary of technical contributions.** We propose GUARD, a novel retention-aware unlearning framework for large language models (LLMs) with the following unique features:

- *Retention-aware objective reweighting.* GUARD chooses unlearning weights inversely proportionally to the retention attribution score of each sample, with a controlled variance that allows for preserving effective forgetting. This mitigates the utility degradation seen in retention-unaware methods.

- *Attribution tailored to LLM unlearning.* GUARD uses a gradient-based retention attribution score that quantifies the "alignment" of a forget sample with retained knowledge which is tailored to unlearning without requiring retraining or computing of the Hessian.

- *Theoretical Guarantees.* We theoretically show that our attribution score lower-bounds conventional influence measures. Additionally, we prove that GUARD improves absolute retention, preserves effective forgetting, and significantly boosts retention efficiency over standard baselines such as Gradient Ascent.

- *Extensive Empirical Validation.* We conduct comprehensive experiments on the TOFU and MUSE benchmarks, covering four non-loss reweighting baselines (GA, GD, KM, PO), their GUARD-augmented counterparts (GA+GUARD, GD+GUARD, KM+GUARD, PO+GUARD), four loss reweighting baselines (NPO, SimNPO, FLAT, SimImP), as well as NPO+GUARD (we selected NPO for integration with GUARD as it demonstrated high stability and strong performance). For all these settings, GUARD consistently enhanced retention while maintaining forgetting performance comparable to baselines, demonstrating its effectiveness as a general framework applicable to diverse unlearning systems.

## 2 LLM UNLEARNING PRELIMINARIES

**Data – The Forget and Retain Sets.** We define a datapoint as $(x, y)$, where $x \in \mathbb{R}^p$ represents the input of the LLM, while $y \in \mathbb{R}^q$ denotes the corresponding ground-truth label or response. The whole dataset used for fine-tuning an LLM is denoted as $D_0 = \{(x_i, y_i)\}_{i \in [n_0]}$, where $n_0$ is the total number of datapoints. The Forget Set, denoted by $D_f$, consists of datapoints requested for removal, while the Retain Set, denoted by $D_r$, contains datapoints that should be kept. The index sets of these datapoints are defined as $I_f := \{i \mid (x_i, y_i) \in D_f\}$ and $I_r := \{j \mid (x_j, y_j) \in D_r\}$. We also let $n_f := |I_f|$ and $n_r := |I_r|$ represent the respective cardinalities of the sets. Clearly, $n_f + n_r = n_0$ and $D_0 = D_f \cup D_r$.

**The empirical loss.** Let $\Theta$ be the parameter space, and let an LLM be parameterized by $\theta \in \Theta$. Given a loss function $l : \mathbb{R}^{p \times q} \mapsto \mathbb{R}$, the loss incurred on a datapoint $(x, y)$ equals $l(x, y; \theta)$. For a dataset $D$, the empirical loss objective is given by:

$$L_D(\theta) = \frac{1}{|D|} \sum_{(x_i, y_i) \in D} l(x_i, y_i; \theta).$$

**Standard LLM unlearning.** *Standard LLM unlearning* refers to the process of adapting an LLM initially trained on $D_0$ in order to remove the influence of a designated Forget Set $D_f$ (see Fig. 2). One typical approach is *Gradient Ascent* (GA), which updates the model parameters $\theta_0$ as follows:

$$\theta_{GA} = \theta_0 + \eta \cdot \nabla_\theta L_{D_f}(\theta_0)$$
$$= \theta_0 + \eta \cdot \sum_{i \in I_f} \frac{1}{n_f} \cdot \nabla_\theta l(x_i, y_i; \theta_0), \text{ where } \eta \text{ denotes the unlearning rate.} \quad (1)$$

Unlearning mechanisms other than Gradient Ascent are discussed in Appendix E.1.

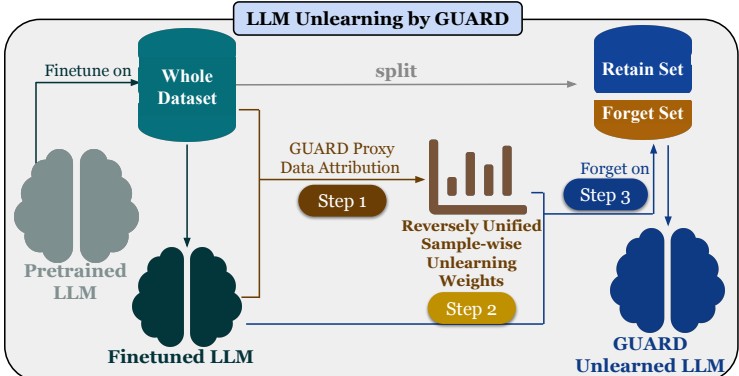

Figure 1: The GUARD pipeline: Step 1) Calculation of proxy data attribution. Step 2) Calculation of unlearning weights based on data attribution, followed by reverse unification of the scores. Step 3) Incorporation of the computed retention-aware weights for reduction of the influence of the Forget Set and preservation of desired knowledge.

## 3 METHOD: GUARD

### 3.1 GUARD PIPELINE

Unlike traditional methods that assign retention-unaware unlearning weights across all data points (e.g., uniform weights as in GA, $\frac{1}{n_f}$; see Eq. 1), GUARD introduces a novel objective function that controllably leverages sample loss weights that are inversely correlated to data attribution scores, so as to model retention (Section 3.4). Additionally, GUARD uses a new data attribution scheme specifically tailored to the LLM unlearning setting (Section 3.2). The GUARD pipeline is outlined in Fig. 1, and it comprises the following steps:

Step 1 *Computing the data attribution scores for retention:* Estimating proxy data attribution scores for all datapoints in the Forget Set, as described in Section 3.2.

Step 2 *Deriving retention-aware unlearning weights:* Calculating unlearning weights that are inversely related to the computed attribution scores, with the degree of adjustment modulated by a temperature parameter $\tau$ (Section 3.3).

Step 3 *Performing GUARD unlearning:* Optimizing the GUARD objective using the retention-aware unlearning weights, as detailed in Section 3.4.

### 3.2 DATA ATTRIBUTION FOR LLM UNLEARNING

Conventional data attribution methods estimate the influence of individual training samples by measuring the change in the model outputs when a sample is removed from the training set. These approaches typically rely on retraining or Hessian-based computations (see Appendix B.2). However, such methods are impractical in the context of LLM unlearning for two key reasons: (1) their computational cost is prohibitive at the operational scale of LLMs; and (2) the objective of unlearning differs fundamentally from standard attribution, i.e., effective unlearning necessitates different performance "shifts," namely, degraded performance on the Forget Set $\mathcal{D}_f$ and improved performance on the Retain Set $\mathcal{D}_r$ (rather than a uniform unidirectional performance change over the entire dataset). To address these challenges, we propose a proxy attribution score $a_i^{\text{GUARD}}$ for a Forget Sample $i$, defined as an inner product of gradients:

$$a_i^{\text{GUARD}} := J_{avg}^{\mathcal{D}_r}(\theta_0)^\top \cdot J_i(\theta_0), \quad \forall i \in I_f, \tag{2}$$

where $J_{avg}^{\mathcal{D}_r}(\theta_0)$ denotes the mean gradient over $\mathcal{D}_r$ and $J_i(\theta_0)$ is the gradient of sample $i$, both evaluated for the initial model parameters $\theta_0$. This formulation captures the alignment between the information content of the Forget and the Retain Set. Representing sample-level influence via gradients is a well-established practice, particularly in LLMs Nguyen et al. (2025); Wang et al. (2025b). In addition to being intuitive and computationally efficient, our proxy attribution score lends itself to theoretical analysis, as described in Appendix C.

### 3.3 Retention-Aware Unlearning Weights

A large retention attribution score for a Forget Sample $i$ indicates that the sample $(x_i, y_i)$ has a strong positive influence on the performance of the model over the Retain Set $\mathcal{D}_r$ (e.g., its removal leads to a large performance drop on $\mathcal{D}_r$). We allocate unlearning weights to data samples inversely proportionally to their attribution scores; at the same time, we modulate the variance of the weight distribution by a temperature-controlled softmax. The unlearning weight for a Forget Sample $i$ equals

$$\omega_i^{\text{GUARD}} := n_f \cdot \frac{e^{-a_i^{\text{GUARD}}/\tau}}{\sum_{j \in I_f} e^{-a_j^{\text{GUARD}}/\tau}}, \quad i \in I_f, \tag{3}$$

where as before, $a_i^{\text{GUARD}}$ is the attribution score defined in Eq. 2, $\tau$ is the temperature hyperparameter, and $n_f = |I_f|$ denotes the size of the Forget Set. The unlearning weights $\omega_i^{\text{GUARD}}$ satisfy the following properties:

*1. Normalization Property,* which asserts that $\frac{1}{n_f} \sum_{i \in I_f} \omega_i^{\text{GUARD}} = 1$.

*2. Inverse Attribution Score Property,* which asserts that unlearning weights should be inversely proportional to the attribution scores of the samples; this mitigates unintended loss of knowledge in $\mathcal{D}_r$ during unlearning of $\mathcal{D}_f$.

*3. Controlled Variance Property,* which asserts that the degree of weight differentiation is governed by a temperature parameter $\tau$ to enable effective forgetting on $\mathcal{D}_f$. Comparable forgetting to baselines is guaranteed theoretically (Section 4) and validated empirically (Section 5).

### 3.4 Unlearning with GUARD

For a given forget dataset $D_f$ and a model parameterized by $\theta$, our novel GUARD unlearning loss objective, $L^{\text{GUARD}} : \mathbb{R}^{p \times q} \to \mathbb{R}$, is defined as

$$L_{D_f}^{\text{GUARD}}(\theta) := \frac{1}{n_f} \sum_{i \in I_f} \omega_i^{\text{GUARD}} \cdot l(x_i, y_i; \theta),$$

where $\omega_i$ is the unlearning weight of datapoint $(x_i, y_i)$ from Section 3.3. Using the GA framework in (Eq. 1) as an example, the model update rule under GUARD becomes:

$$\theta_{\text{GA}}^{\text{GUARD}} = \theta_0 + \frac{\eta}{n_f} \sum_{i \in I_f} \omega_i^{\text{GUARD}} \cdot \nabla_\theta \left[ l(x_i, y_i; \theta_0) \right]. \tag{4}$$

As discussed in Section 3.3, such an unlearning weight allocation (Eq. 4) mitigates unintended forgetting of retained knowledge while it also ensures effective forgetting of undesired information (GUARD objectives for methods other than Gradient Ascent are described in Appendix E.2 and Appendix E.3).

**The GUARD algorithm.** The descriptions in Sections 3.1–3.4 are summarized in Algorithm 1, which outlines the implementation of GUARD within the GA unlearning framework (Eq.1). We adopt the standard unlearning protocol with a single forget epoch Yao et al. (2024a), although the algorithm naturally generalizes to multiple unlearning epochs as needed.

---

**Algorithm 1** GUARD: Guided Unlearning and Retention via Data Attribution

---

1: **Input**: Initial model weights $\theta_0$; unlearning rate $\eta$; temperature $\tau$.
2: Compute retention attribution $a_i^{\text{GUARD}}$ for all $i \in I_f$ using Eq. 2        ▷ Prepare Attribution
3: Compute unlearning weights $\omega_i$ for all $i \in I_f$ using Eq. 3     ▷ Prepare Sample-wise Weights
4: Update model $\theta_0$ to $\theta_{\text{GA}}^{\text{GUARD}}$ by Eq. 4                 ▷ Update Model
5: **Output**: Unlearned model weights $\theta_{\text{GA}}^{\text{GUARD}}$.

---

## 4 Theoretical Guarantees

Let $\psi \in \{0, \text{GA}, \text{GUARD}\}$ denote the superscripts corresponding to models trained without unlearning, trained with standard/baseline unlearning (GA), and GUARD, respectively. For a model

parameterized by $\theta^\psi$, we define the empirical loss over the Forget Set $\mathcal{D}_f$ and the Retain Set $\mathcal{D}_r$ as:

$$L_{\mathcal{D}_f}^\psi = \frac{1}{n_f} \sum_{i \in I_f} l(x_i, y_i; \theta^\psi), \;\; L_{\mathcal{D}_r}^\psi = \frac{1}{n_r} \sum_{i \in I_r} l(x_i, y_i; \theta^\psi), \;\; \text{where } n_f = |I_f|, \, n_r = |I_r|.$$

Let $\overline{g}_f := \frac{1}{n_f} \sum_{i \in I_f} g_i$ and $\overline{g}_r := \frac{1}{n_r} \sum_{i \in I_r} g_i$ denote the average gradients over the Forget and Retain Sets, respectively. We denote the global alignment between the two sets as $\kappa := \langle \overline{g}_f, \overline{g}_r \rangle$ and introduce a normalized alignment metric $\delta_\kappa := \frac{\kappa}{\tau}$, where $\tau$ is the temperature hyperparameter in the GUARD multiplier. Denote the maximum of the norm of model updates upon unlearning as $\delta_\theta := \max_{\psi \in \{\text{GA,GUARD}\}} \|\theta^\psi - \theta^0\|_2$, and define $\delta := \delta_\kappa + \delta_\theta$. For each Forget Sample index $j \in I_f$, we have $\kappa_j := \langle \overline{g}_r, g_j \rangle$, whose variance equals

$$\sigma_\kappa^2 := \frac{1}{n_f} \sum_{j \in I_f} \kappa_j^2 - \left( \frac{1}{n_f} \sum_{j \in I_f} \kappa_j \right)^2.$$

**Evaluation metric**. We define the *Sacrifice Rate* $\rho^\psi$ to quantify the performance loss on the Retain Set per "unit" of performance loss on the Forget Set,

$$\rho^\psi := \frac{L_{\mathcal{D}_r}^\psi - L_{\mathcal{D}_r}^0}{L_{\mathcal{D}_f}^\psi - L_{\mathcal{D}_f}^0}.$$

A smaller value of $\rho^\psi$ is indicative of better knowledge retention.

**Assumption 1** (Knowledge entanglement between the Forget and Retain Sets). *Assume that the following inequalities hold: (1) $\langle \overline{g}_r, \overline{g}_f \rangle > 0$; (2) $\langle \overline{g}_r, g_i \rangle / \tau \ll 1$, $\forall i \in I_f$.*

Next, note that gradients can be seen as proxies for information flow Nguyen et al. (2025); Wang et al. (2025b). Consequently, condition (1) implies a global alignment between the Forget and Retain Sets, implying that their entanglement—unlearning $\mathcal{D}_f$ typically harms the performance on $\mathcal{D}_r$. Condition (2) assumes weak alignment between each Forget Sample and the Retain Set, enabling selective unlearning without significantly compromising the retained knowledge. The two inequalities jointly yield to $0 < \delta_\kappa = \kappa/\tau \ll 1$.

**Assumption 2** (Small updates). *We assume that the second and higher powers of $\delta_\theta$, the maximal norm of model weight changes upon unlearning, are negligible (and therefore omitted in the analysis).*

The above assumption is commonly used in influence-function-based data attribution Ellis et al. (2022), and is appropriate whenever unlearning introduces small perturbations in the model weights.

**Assumption 3** (Isotropic gradients). *Assume the gradients in $I_f$ are approximately isotropic. Formally, let $\Sigma_f := \frac{1}{n_f} \sum_{j \in I_f} g_j g_j^\top$ denote the empirical second-moment matrix of the gradients of $I_f$. Then, $\Sigma_f \approx \lambda I$ for some scalar $\lambda > 0$.*

Similar isotropy assumptions were adopted in prior data attribution works for approximating gradient covariance matrices in high-dimensional neural networks Ingram et al. (2023).

We show next that GUARD significantly reduces the Sacrifice Rate (Theorem 3) by preserving performance on the Retain Set (Lemma 1) while effectively unlearning the Forget Set (Lemma 2).

**Lemma 1** (Retain loss reduction by GUARD). *Under Assumptions 1-3, the loss on the Retain Set under GUARD is lower than that under GA, i.e.,*

$$L_{\mathcal{D}_r}^{GA} - L_{\mathcal{D}_r}^{GUARD} = \frac{\eta}{(1 - \delta_\kappa)\tau} \cdot \sigma_\kappa^2 + \mathcal{O}(\delta^2), \; \text{where } \eta \text{ denotes the unlearning rate.}$$

The proof is deferred to Appendix D. Lemma 1 shows that the improvement offered by GUARD is proportional to the alignment variance $\sigma_\kappa^2$ and that it scales with $\frac{\eta}{(1-\delta_\kappa)\tau}$. A large variance suggests strong interactions between the Forget Samples and the Retain Set, which GUARD leverages to reduce the loss. Retention improves with increasing the unlearning rate $\eta$ and decreasing the temperature $\tau$, as long as $\delta_\kappa := \kappa/\tau \ll 1$.

**Lemma 2** (The forget loss of GUARD is comparable to that of GA). *Under Assumptions 1 and 2, the difference in forget loss between GUARD and GA is bounded by*

$$\left| L_{\mathcal{D}_f}^{GA} - L_{\mathcal{D}_f}^{GUARD} \right| = \delta_\kappa \eta \cdot \|\overline{g}_f\|_2^2 + \mathcal{O}(\delta^2).$$

The proof is deferred to Appendix D. Lemma 2 confirms that GUARD remains effective at unlearning, with a forget loss comparable to that of GA. The gap is small, controlled by $\delta_\kappa \ll 1$, and can be further reduced by a smaller squared averaged forget gradient norm $\|\overline{g}_f\|_2^2$ and unlearning rate $\eta$.

**Theorem 3** (Sacrifice rate reduction by GUARD). *Under Assumptions 1-3, GUARD can be shown to reduce the Sacrifice Rate relative to GA as*

$$\rho^{GA} - \rho^{GUARD} = \frac{\kappa^2 + \sigma_\kappa^2}{\tau \cdot \|\overline{g}_f\|_2^2} + \mathcal{O}(\delta^2).$$

The proof is deferred to Appendix D. The reduction is large when 1) the global alignment $\kappa$ is large; 2) the variance $\sigma_\kappa^2$ is large; 3) the average forget gradient norm is small; and 4) the temperature $\tau$ is small (provided that $\delta_\kappa := \kappa/\tau \ll 1$).

Using Lemma 1, Lemma 2, and Theorem 3, in Appendix D.1 we provide an in-depth analysis of how GUARD unlearning is influenced by both tunable hyperparameters (the unlearning rate $\eta$ and temperature $\tau$) and intrinsic characteristics of the data/model (knowledge alignment $\kappa$, alignment variance $\sigma_\kappa^2$, and average forget gradient norm $\|\overline{g}_f\|$).

## 5 EXPERIMENTS

**The dataset.** We evaluate GUARD on **TOFU** Maini et al. (2024b) and **MUSE** Shi et al. (2024a) benchmarks for LLM unlearning Jia et al. (2024); Ji et al. (2024). TOFU comprises $4,000$ Q-A pairs from 200 synthetic author profiles. The data is split into: a) *Forget/Retain Sets*, sampled to unlearn $1\%$, $5\%$, or $10\%$ of the data; b) *Generalization Sets* (*Real-Author*, *Real-World-Facts*) to evaluate downstream retention and generalization. MUSE provides two real-world corpora: a) *NEWS (BBC articles)* with 0.8M/1.6M token Forget/Retain Sets; b) *BOOKS* with Harry Potter books as a 1.1M token Forget Set, and FanWiki as 0.5M token Retain Set. Each corpus includes verbatim text and knowledge sets with question-answer pairs derived from the original content.

**The backbone LLM.** Following the original setups of the benchmarks, we evaluate unlearning performance using *Phi-1.5B* Li et al. (2023) and *Llama-2-7B* Touvron et al. (2023) on TOFU, *Llama-2-7B* on MUSE NEWS corpus, and *ICLM-7B* Shi et al. (2024b) on MUSE BOOKS corpus.

**Baselines.** We adopt four non-loss reweighting unlearning baselines provided by TOFU and implement GUARD on them. These include: *Gradient Ascent (GA)* Thudi et al. (2022), *Gradient Difference (GD)* Liu et al. (2022), *KL Minimization (KM)* Chundawat et al. (2023), and *Preference Optimization (PO)* Maini & et al. (2024). Detailed reviews of these baselines and corresponding GUARD algorithms are available in Appendix E.1 and E.3. It is worth noting that since MUSE is a textual dataset and PO is only applicable to QA-Pair datasets, we do not report PO results for MUSE. Additionally, we include results for four recent loss reweighting methods: *Negative Preference Optimization (NPO)* Zhang et al. (2024a), *SimNPO* Fan et al. (2025), *FLAT* Wang et al. (2024b) (we use the TV variant as f-divergence required by FLAT since it achieves the best empirical performance) and *SatImp* Yang et al. (2025). For fairness of comparisons, we *apply GUARD to NPO* and compare it against the four loss reweighting methods. We chose NPO as the base method due to its demonstrated stability and excellent performance. However, it is important to note that GUARD can be adapted with equal ease to all other loss reweighting methods to provide further improvements if needed. Details regarding the baselines are provided in Appendix B.

**Evaluation metrics.** We use tailor-made metrics for TOFU and MUSE benchmarks due to their distinct evaluation criteria.

TOFU Evaluations: To measure forgetting, retention, and the trade-off between the two, we evaluate the Absolute Performance and Sacrifice Rate.
(a) *Absolute Performance*. Given a dataset $D$ and model $\theta$, we use the Absolute Performance assessment $\epsilon(D; \theta)$ from the TOFU benchmark (e.g., ROUGE-L, Probability, Truth Ratio). A lower value of $\epsilon(\mathcal{D}_f; \theta)$ indicates better forgetting, while a higher value of $\epsilon(\mathcal{D}_r; \theta)$ indicates better retention.
(b) *Sacrifice Rate.* We introduce the *Sacrifice Rate*, denoted as $\rho$, and defined as

$$\rho_\epsilon(D) = \frac{\epsilon(D_r; \theta^0) - \epsilon(D_r; \theta)}{\epsilon(D_f; \theta^0) - \epsilon(D_f; \theta)}. \tag{5}$$

This metric measures the relative performance degradation on the desired sets per unit of performance decrease on the Forget Set. The lower $\rho_\epsilon$, the better relative knowledge retention.

MUSE Evaluations: Following the original MUSE framework, we evaluate methods according to four key criteria: (a) *Forget Set VerbMem.* This criteria measures exact text reproduction from the Forget Set. Lower values indicate better unlearning. (b) *Forget Set KnowMem.* This metric assesses the ability of the model to answer questions about the Forget Set. Lower values indicate better unlearning. (c) *PrivLeak.* This metric quantifies membership inference vulnerability. Values within [-5%, 5%] indicate successful unlearning, while values outside this range suggest over/under-unlearning. (d) *Retain Set KnowMem.* This criteria is used to evaluate knowledge retention on the Retain Set. Higher values of the score indicate better utility preservation.

**Configuration and hyperparameters.** For details, refer to Appendix F.1.

**Results.** We evaluate the effectiveness of GUARD by focusing on the LLaMA-2-7B backbone, and defer the results for Phi-1.5B to Appendix F.3, as both models exhibit similar behaviors. We report on experimental results for both TOFU (Sacrifice Rates in Table 1 and Absolute Performance metrics in Table 2) and MUSE (four criteria in Table 3). Due to space limitations, in the main text we focus on results for 10% unlearning on TOFU as this is the most challenging scenario, while results for 1% and 5% data unlearning are delegated to Appendix F.2. For the same reason, we only present results for MUSE NEWS and delegate the MUSE BOOKS dataset analyses to Appendix F.4.

Table 1: Unlearning experiments with LLaMA-2-7B on 10% of all data points in TOFU. Across all baselines, GUARD markedly mitigates retention degradation, with the most substantial improvement observed on the Retain Set where it reduces $\rho_t$ by up to 194.92% relative to GA.

| Methods | Retain Set($\downarrow$) | | | Real Author Set($\downarrow$) | | | Real World Set($\downarrow$) | | |
|---|---|---|---|---|---|---|---|---|---|
| | $\rho_r$ | $\rho_p$ | $\rho_t$ | $\rho_r$ | $\rho_p$ | $\rho_t$ | $\rho_r$ | $\rho_p$ | $\rho_t$ |
| GA | 102.57 | 124.08 | 421.13 | 80.47 | 111.00 | 371.88 | 87.06 | 79.70 | 246.03 |
| GA + GUARD | **68.46** | **84.76** | **226.21** | **64.67** | **54.82** | **181.30** | **73.90** | **37.31** | **157.97** |
| KM | 101.19 | 112.03 | 410.22 | 79.48 | 55.66 | 271.93 | 86.14 | 62.60 | 126.89 |
| KM + GUARD | **67.63** | **80.05** | **218.88** | **63.96** | **36.16** | **117.02** | **75.75** | **34.60** | **81.21** |
| PO | 89.88 | 104.47 | 221.31 | 44.00 | 45.73 | 171.55 | 8.38 | 37.60 | 28.27 |
| PO + GUARD | **62.73** | **79.62** | **124.23** | **29.04** | **30.39** | **126.27** | **-7.20** | **23.49** | **-7.21** |
| GD | 86.37 | 94.12 | 117.88 | 38.49 | 15.74 | 60.67 | -3.05 | 8.77 | -20.38 |
| GD + GUARD | **59.94** | **76.89** | **106.81** | **25.91** | **3.28** | **13.24** | **-12.53** | **-2.27** | **-22.83** |
| NPO | 85.58 | 106.24 | 201.33 | 42.91 | 42.47 | 101.43 | 8.11 | 33.24 | 20.32 |
| SimNPO | 76.12 | 91.22 | 151.51 | 36.11 | 32.94 | 52.32 | 5.83 | 13.51 | 8.42 |
| FLAT (TV) | 73.11 | 85.46 | 158.63 | 29.32 | 11.27 | 41.33 | 5.30 | 6.26 | -12.97 |
| SatImp | 71.93 | 83.11 | 131.75 | 31.08 | 15.94 | 53.28 | 8.64 | 8.85 | -5.62 |
| NPO + GUARD | **61.12** | **78.39** | **118.47** | **23.88** | **4.24** | **11.62** | **-7.87** | **-4.55** | **-13.69** |

**Effectiveness of GUARD on TOFU.** As shown in Table 1, GUARD consistently offers lower Sacrifice Rates across all datasets and baselines, indicating improved retention of unforgotten knowledge. In particular, GUARD reduces the Sacrifice Rate by up to 194.92% in Truth Ratio on the Retain Set when unlearning 10% of training data. From Table 2, we see that GUARD also significantly improves Absolute Performance on the Retain Set while maintaining comparable performance to baselines on the Forget Set, demonstrating its ability to enhance retention without compromising forgetting. Notably, GUARD increases the Truth Ratio by up to 32.08% over the GA baseline on the Retain Set while maintaining similar degradation levels on the Forget Set as the standard method.

Additionally, GUARD applied on NPO significantly outperforms NPO, SimNPO, FLAT (TV) and SatImp across all three evaluation sets. Specifically, NPO + GUARD achieves substantially lower Sacrifice Rates than SatImp, with reductions of up to 19.18% in Truth Ratio on the Retain Set. Also, NPO + GUARD increases the Truth Ratio by up to 32.08% over SatImp on the Retain Set while retaining similar degradation levels on the Forget Set. This demonstrates that our attribution-driven approach provides superior knowledge retention compared to existing loss reweighting strategies.

**Effectiveness of GUARD on MUSE.** Based on Table 3, GUARD demonstrates similar effectiveness on MUSE as observed on TOFU. GUARD consistently improves retention performance across all baselines while maintaining effective forgetting, with NPO + GUARD achieving the best overall performance on Retain Set KnowMem (45.9) compared to all competing loss reweighting methods. We note that GUARD exhibits somewhat higher PrivLeak scores, which reflects the fact that retention of relevant knowledge naturally makes the model more vulnerable to membership inference attacks. Moreover, this moderate privacy degradation (10.5% increase for NPO + GUARD) is substantially

Table 2: Impact of GUARD on Absolute Performance evaluated over the Retain and Forget Sets. Arrows (↑/↓) indicate whether higher/lower values are preferred. GUARD consistently improves performance on the Retain Set and maintains performance on the Forget Set relative to baselines.

| Methods | Retain Set (↑) | | | Forget Set (↓) | | |
|---------|---------|-------|---------|---------|-------|---------|
| | ROUGE-L | Prob. | T. Ratio | ROUGE-L | Prob. | T. Ratio |
| w/o Unlearn | 99.35 | 84.36 | 98.20 | 98.99 | 84.08 | 96.27 |
| GA | 48.81 | 41.97 | 19.32 | **43.18** | 48.35 | 76.60 |
| GA + GUARD | **63.63** | **52.83** | **51.40** | 43.44 | **48.23** | **76.55** |
| KM | 51.05 | 44.62 | 26.21 | 42.91 | 48.76 | 74.92 |
| KM + GUARD | **65.36** | **53.12** | **53.89** | **42.90** | **48.09** | **74.33** |
| PO | 68.87 | 46.25 | 39.83 | 35.44 | 47.49 | 67.17 |
| PO + GUARD | **83.22** | **54.11** | **63.09** | **35.05** | **47.37** | **66.73** |
| GD | 72.50 | 52.25 | 61.16 | **33.45** | **43.68** | 63.85 |
| GD + GUARD | **87.15** | **60.47** | **70.20** | 33.76 | 43.80 | **63.78** |
| NPO | 70.01 | 48.30 | 55.26 | 32.12 | 42.71 | 63.89 |
| SimNPO | 73.12 | 53.40 | 60.28 | 32.97 | **42.06** | 62.58 |
| FLAT (TV) | 68.72 | 50.21 | 58.93 | 37.44 | 46.56 | 66.91 |
| SatImp | 68.60 | 49.51 | 60.36 | **31.38** | 42.25 | **62.22** |
| NPO + GUARD | **86.25** | **60.54** | **71.94** | 31.79 | 42.57 | 63.26 |

outweighed by the significant utility gains in knowledge retention (a 27.8% improvement on the Retain Set KnowMem) and highly improved forgetting performance (a 22.8% reduction on Forget Set KnowMem), demonstrating that the retention benefits far exceed the privacy loss.

**Additional results.** 1) **Computational efficiency analysis** (Appendix F.5): We present a runtime comparisons demonstrating that GUARD introduces minimal computational overhead, with data attribution representing a one-time preprocessing step that amortizes across multiple unlearning operations. 2) **Visualization of proxy attribution and reallocated unlearning power** (Appendix F.6): We provide visualizations that confirm the intuition that GUARD reallocates unlearning power based on the proxy attribution scores to mitigate retention losses. 3) **Hyperparameter sensitivity analysis** (Appendix F.7): We analyze the sensitivity of the temperature parameter $\tau$, supporting our claim that tuning $\tau$ controls the quality of retention.

Table 3: Unlearning experiments with LLaMA-2-7B on MUSE NEWS dataset.

| Methods | Forget Set VerbMem (↓) | Forget Set KnowMem (↓) | PrivLeak ($\in [-5\%, 5\%]$) | Retain Set KnowMem (↑) |
|---------|---------|---------|---------|---------|
| | | NEWS | | |
| w/o Unlearn | 58.4 | 63.9 | −99.8 | 55.2 |
| GA | 0.0 | 0.0 | **5.8** | 0.0 |
| GA + GUARD | **0.0** | **0.0** | 6.5 | **8.4** |
| KM | **28.1** | 50.3 | −95.4 | 43.5 |
| KM + GUARD | 28.5 | **49.7** | −101.2 | **50.1** |
| GD | 4.4 | 32.4 | **101.7** | 27.6 |
| GD + GUARD | **2.62** | **25.9** | 106.8 | **45.1** |
| NPO | 2.7 | 56.8 | 87.7 | 35.9 |
| SimNPO | 2.2 | 45.1 | 88.9 | 38.6 |
| FLAT (TV) | 2.3 | **42.6** | **81.2** | 30.3 |
| SimImP | 2.5 | 46.0 | 92.4 | 39.5 |
| NPO + GUARD | **2.2** | 43.8 | 96.9 | **45.9** |

## 6 CONCLUSION

We present GUARD, a guided unlearning framework that leverages data attribution to mitigate unintended forgetting in LLM unlearning. By adapting unlearning strength based on data influence, GUARD preserves essential knowledge while removing target data. Our theoretical analysis and experiments demonstrate substantial improvements over existing baselines across LLM architectures.

## ETHICS STATEMENT

The authors acknowledge adherence to the ICLR Code of Ethics and recognize that this work addresses legitimate privacy concerns by developing methods to selectively remove sensitive information from large language models in support of data protection regulations. We honestly report that current unlearning methods, including GUARD, may not provide absolute guarantees of information removal, and practitioners should carefully evaluate models for unintended bias or fairness implications after unlearning procedures. All experiments were conducted using established benchmarks with proper attribution and transparent reporting of results and limitations. This research contributes to making AI systems more controllable and privacy-preserving, though we recognize that responsible deployment requires interdisciplinary collaboration with legal, policy, and ethics experts. The authors commit to sharing code and experimental details to support reproducible research while being mindful of potential misuse scenarios.

## REPRODUCIBILITY STATEMENT

To ensure reproducibility of our results, we have made comprehensive efforts to document all experimental details and follow established evaluation protocols. Our experiments strictly adhere to the evaluation frameworks and protocols specified in the original TOFU (Maini et al., 2024b) and MUSE (Shi et al., 2024a) benchmark papers, and we encourage researchers to follow the guidance provided in those publications for proper reproduction of baseline comparisons. All hyperparameter settings, model configurations, and experimental setup details for the GUARD method are provided in Appendix F.1, including learning rates, batch sizes, temperature parameters, and hardware specifications used across different model architectures. The algorithmic implementation of GUARD is detailed in Algorithm 1 and Algorithm 2, with mathematical formulations provided in Section 3 and theoretical derivations included in Appendix D. Our evaluation metrics and experimental protocols for both absolute performance and sacrifice rate measurements are comprehensively described in Section 5, with additional results across different data splits and model architectures provided in Appendices F.2 through F.4. We commit to releasing our implementation code and detailed experimental scripts to facilitate exact reproduction of our reported results.

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

# Appendix

CONTENTS

## A    LLM USAGE STATEMENT

We used a large language model (LLM) *Claude Sonet 4* for language polishing of parts of the manuscript (phrasing, grammar, and clarity). The LLM was only used to improve presentation; it did not contribute to the technical content, experimental design, data analysis, or conclusions. All substantive scientific claims, results, and interpretations are the authors' own, and the authors take full responsibility for the final text. Any LLM-generated text was reviewed, edited, and approved by the authors.

## B    RELATED WORKS

### B.1    UNLEARNING FOR LLM

Existing LLM Unlearning approaches broadly fall into two categories: Fine-tuning-based and inference-time-based methods. Fine-tuning approaches such as Gradient Ascent (GA) Yao & et al. (2023), Preference Optimization Maini & et al. (2024) and variants thereof, modify model weights by adjusting training objectives, typically through loss maximization on the forget set Zhang & et al. (2024a); Li & et al. (2024) and incorporation of regularization terms based on KL divergence, retained data consistency, or prompt-guided responses Shi & et al. (2024); Choi & et al. (2024); Wang et al. (2024a). Recent approaches such as WAGLE Jia et al. (2024) improve computational efficiency by isolating the parameters most relevant to forgetting. However, fine-tuning-based methods frequently suffer from *excessive forgetting* and *retention degradation* Zhang & et al. (2024b), compromising the utility of the retained model knowledge. A complementary line of work explores inference-time unlearning Pawelczyk & et al. (2023); Huang & et al. (2024); Thaker & et al. (2024), which avoids model updates by modifying prompts or logits at inference to suppress specific outputs Liu et al. (2024); Ji et al. (2024). While computationally attractive, these methods do not guarantee actual forgetting, as sensitive information remains embedded in the unchanged model parameters. Emerging directions Yao et al. (2024b); Zhang et al. (2024b) also investigate properties beyond forgetting and retention, such as robustness of unlearning mechanisms and recovery from catastrophic forgetting.

**Fine-Tuning-Based Methods.** Many unlearning techniques modify model weights through loss-based fine-tuning strategies, such as GA Yao & et al. (2023), preference optimization (PO) Maini & et al. (2024), negative preference optimization (NPO) Zhang & et al. (2024a), and representation misdirection Li & et al. (2024). These methods typically maximize the loss on the forget set while incorporating regularization objectives such as KL divergence minimization, joint training with retain data Shi & et al. (2024), or prompt-based desirable loss surrogates Choi & et al. (2024). Despite their effectiveness in forgetting, these methods often degrade performance on retained knowledge due to excessive forgetting Zhang & et al. (2024b). WAGLE Jia et al. (2024) focuses on improving computation efficiency by isolating critical parameters for forgetting via weight attribution, whereas Wang et al. (2024a) adds a regularization term to penalize low-quality generations, yet both empirically fail to preserve retention. Other fine-tuning efforts include Eldan & Russinovich (2023); Patil & et al. (2024); Jia & et al. (2024), but none achieve robust retention preservation.

**Input-Based and In-Context Methods.** Input-level methods operate at inference time by manipulating inputs to suppress undesired outputs without altering model weights. These include prompt filtering Liu et al. (2024), instruction-based tuning Pawelczyk & et al. (2023), and logit-based interventions Huang & et al. (2024); Ji et al. (2024). ECO Prompts Liu et al. (2024) use prompt classifiers and zeroth-order optimization to induce forgetting, while Unlearning from Logit Difference (ULD) Ji et al. (2024) employs an assistant LLM to reduce side effects. Although computationally efficient, these methods rely on access to sensitive data at inference time, which limits their privacy guarantees Thaker & et al. (2024); Liu & et al. (2024).

**Alternative Directions.** Recent efforts explore complementary directions beyond weight-based or input-level interventions. Multimodal unlearning Li et al. (2024a), adversarial robustness Schwinn et al. (2024), and standardized benchmarks Yao et al. (2024b); Wang et al. (2025a) enhance evaluation protocols but do not directly address retention. FLAT Wang et al. (2024b) eliminates the need for Retain Data by maximizing the f-divergence between template and forget responses, while Yang et al. Yang et al. (2025) analyze loss reweighting mechanisms through saturation and importance-based categorizations. Negative Preference Optimization (NPO) Zhang et al. (2024a) treats Forget Data as negative responses in preference optimization, providing bounded objectives with adaptive gradient smoothing to prevent catastrophic collapse during unlearning. SimNPO Fan et al. (2025) further addresses reference model bias in NPO by adopting a reference-free, length-normalized approach that more effectively allocates optimization power across forget samples with varying difficulty levels. Utility-guided methods, such as gradient-based influence modeling Wang et al. (2025b), propose retention-aware metrics without offering actionable strategies. Other approaches like quantization-based recovery Zhang et al. (2024b) attempt to restore forgotten knowledge post hoc but remain orthogonal to the goal of preserving retention during unlearning.

### B.2 DATA ATTRIBUTION

Data attribution in machine learning aims to quantify the influence of individual data points on model behavior, supporting tasks such as debugging, data cleaning, and interpretability. Classical approaches, such as influence functions, estimate the effect of removing a single data point on model parameters or predictions. Koh & Liang (2017) adapted influence functions to deep learning, illustrating their value in identifying mislabeled data and interpreting model decisions. However, subsequent studies revealed notable limitations. For example, Basu et al. (2021) showed that influence estimates are highly sensitive to hyperparameters and model architecture, particularly in deep networks. Han & Liu (2022) further found that these methods can fail basic sanity checks, attributing high influence to unrelated data. Bae et al. (2022) also demonstrated that influence functions may estimate unintended quantities in deep neural networks, raising concerns about their reliability in complex models.

To address the limitations of classical influence functions, recent work has introduced more robust and scalable data attribution techniques. Among these, direct estimators such as regression-based datamodels have demonstrated improved predictive accuracy and generalization, particularly in large-scale settings where computational resources are sufficient Ellis et al. (2022). Two state-of-the-art approaches—TRAK Ellis et al. (2022) and EK-FAC Ingram et al. (2023)—build upon this paradigm and offer practical solutions across diverse modalities. TRAK approximates a deep neural network with a linear surrogate model and estimates influence by ensembling over multiple model checkpoints and random projections. EK-FAC improves efficiency through a combination of strategies: partitioning the Hessian into block-diagonal components, expressing gradients as Kronecker products of smaller factors, and applying spectral regularization to stabilize curvature estimates.

Despite their empirical success, these methods remain computationally prohibitive for large-scale models such as large language models (LLMs). Moreover, their underlying attribution philosophy is not directly aligned with the goals of machine unlearning, which demands targeted forgetting while preserving non-targeted knowledge.

## C THEORETICAL SOUNDNESS OF OUR DATA ATTRIBUTION TAILORED FOR LLM UNLEARNING

### C.1 PRELIMINIARIES: CONVENTIONAL DATA ATTRIBUTION

**Leave-One-Out Framework.** Data attribution quantifies the impact of a specific training datapoint on the model's loss under the Leave-One-Out framework Rad & Maleki (2018). Given a dataset $D$, the influence of a datapoint $(x_j, y_j)$ on the model's optimal loss is defined as:

$$\Delta_L^j = \min_\theta L_D(\theta) - \min_\theta L_{D \setminus \{(x_j, y_j)\}}(\theta).$$

**Influence Function.** The *Influence Function* Bae et al. (2022) provides an efficient approximation for the leave-one-out influence. Let $H_D(\theta)$ denote the Hessian matrix summing over data-wise

second derivatives: $H_D(\theta) := \sum_{(x_i, y_i) \in D} \nabla^2 l(x_i, y_i; \theta)$, The influence function for a datapoint $(x_j, y_j)$ is then given by:

$$a_j^{\text{IF}} = J_{avg}(\theta_0)^\top \cdot [H_D(\theta_0)]^{-1} \cdot J_j(\theta_0), \tag{6}$$

where $J_j(\theta) := \nabla_\theta l(x_j, y_j; \theta)$ represents the Jacobian of the loss function with respect to the model parameters.

## C.2 GUARD ATTRIBUTION

**Key Notations Related to Proxy Attribution**. Let the eigenvalues and eigenvectors of the inverse Hessian matrix $H_D(\theta_0)^{-1}$ be denoted by $\{\lambda_k\}_k$ and $\{v_k\}_k$, respectively. Let $\lambda_{\max}$ and $\lambda_{\min}$ represent the maximum and minimum eigenvalues, respectively. We define the projections of the averaged Jacobian and the datapoint-specific Jacobian onto the eigenbasis as follows:

$$s_{j,k} := J_j(\theta_0) \cdot \frac{v_k}{\|v_k\|}, \quad s_{\text{avg},k} := J_{\text{avg}}(\theta_0) \cdot \frac{v_k}{\|v_k\|}.$$

Additionally, we define $q_k$ as the product of the component-wise decompositions of $J_{\text{avg}}(\theta_0)$ and $J_j(\theta_0)$ in the eigenbasis, i.e., $q_k := s_{\text{avg},k} \cdot s_{j,k} \|v_k\|^2$.

Based on these notations, we propose Assumptions 4 , under which we derive a bound for $a_j^{\text{IF}}$ as defined in Eq. 6, as stated in Lemma 4

**Assumption 4** (Smoothness of the Loss Function). *The loss function $l(x_i, y_i; \theta)$ is twice differentiable with respect to $\theta$, and the second derivatives of the loss function are bounded for all data points.*

*Remark* 1. Many commonly used loss functions, such as the squared loss and logistic loss, satisfy this assumption. This assumption ensures that the Hessian matrix exists and is well-behaved, allowing for stable second-order optimization techniques such as Newton's method.

**Lemma 4** (Bounds on the Influence Function by Proxy Attribution). *Suppose that Assumptions 4 holds. Define $q^+ := \sum_k [\mathbf{1}_{q_k \geq 0} \cdot q_k]$, and $q^- := \sum_k [\mathbf{1}_{q_k < 0} \cdot q_k]$. The influence function estimator $a_j^{IF}$, as defined in Eq. 6, satisfies the following bound:*

$$\lambda_l \cdot a_j^{GUARD} \leq a_j^{IF} \leq \lambda_u \cdot a_j^{GUARD}, \tag{7}$$

*where $\lambda_u := \frac{\lambda_{\max} q^+ + \lambda_{\min} q^-}{q^+ + q^-}$, and $\lambda_l := \frac{\lambda_{\min} q^+ + \lambda_{\max} q^-}{q^+ + q^-}$.*

*Remark* 2. The proof is provided below. As established in Lemma 4, the influence function $a_j^{\text{IF}}$ is scaled by the GUARD proxy attribution $a_j$. Building on this intuition, we later approximate $a_j^{\text{IF}}$ by $C \cdot a_j^{\text{GUARD}}$, where $C$ is a constant that will be canceled by unification (See Section 3.1). A discussion on $C$ is deferred to Appendix C.3. Note that the scale constant $C$ is cancelled by obtaining GUARD multiplier using Eq. 3. Therefore, in practical implementation, we adopt the unscaled proxy attribution:

$$\tilde{a}_i = [J_{avg}^{D_r}]^\top \cdot J_i(\theta_0), \quad \forall i \in I_f \tag{8}$$

*Proof.* We begin by recalling the expression for the influence function:

$$a_j^{\text{IF}} = J_{\text{avg}}(\theta_0)^\top \cdot [H_D(\theta_0)]^{-1} \cdot J_j(\theta_0),$$

where $J_{\text{avg}}(\theta_0)$ is the average Jacobian of the loss function over all data points.

Recall the definitions

$$s_{j,k} := J_j(\theta_0) \cdot \frac{v_k}{\|v_k\|}, \quad s_{\text{avg},k} := J_{\text{avg}}(\theta_0) \cdot \frac{v_k}{\|v_k\|}.$$

We have $J_{avg}(\theta_0) = \sum_k s_{avg,k} \cdot v_k$ and $J_j(\theta_0) = \sum_k s_{j,k} \cdot v_k$. Therefore,

$$a_j^{\text{IF}} = J_{\text{avg}}(\theta_0)^\top \cdot [H_D(\theta_0)]^{-1} \cdot J_j(\theta_0)$$

$$= J_{\text{avg}}(\theta_0)^\top \cdot [H_D(\theta_0)]^{-1} \cdot \sum_k [s_{j,k} \cdot v_k]$$

$$= \left[ \sum_k [s_{avg,k} \cdot v_k] \right]^\top \cdot \sum_k [s_{j,k} \cdot \lambda_k \cdot v_k]$$

As $H_D(\theta_0)$ is symmetric, we have that $H_D(\theta_0)^{-1}$ is asymmetric, and therefore, the eigenvectors of $H_D(\theta_0)^{-1}$ are orthogonal to each other, i.e., $v_i \cdot v_j = 0, \forall i \neq j$. Consequently,

$$
\begin{aligned}
a_j^{\text{IF}} &= \sum_k \lambda_k \cdot s_{avg,k} \cdot s_{j,k} \cdot \|v_k\|_2^2 \\
&= \sum_k \lambda_k \cdot q_k \\
&\leq \sum_k [\mathbf{1}_{q_k \geq 0} \cdot \lambda_{max} \cdot q_k + \mathbf{1}_{q_k < 0} \cdot \lambda_{min} \cdot q_k] \\
&= \sum_k [\mathbf{1}_{q_k \geq 0} \cdot \lambda_u \cdot q_k + \mathbf{1}_{q_k < 0} \cdot \lambda_u \cdot q_k] \\
&= \lambda_u \cdot J_{avg}(\theta_0)^\top \cdot J_j(\theta_0) \\
&= \lambda_u \cdot a_j,
\end{aligned}
$$

which is the upper bound in Lemma 4. Similarly,

$$
\begin{aligned}
a_j^{\text{IF}} &= \sum_k \lambda_k \cdot s_{avg,k} \cdot s_{j,k} \cdot \|v_k\|_2^2 \\
&= \sum_k \lambda_k \cdot q_k \\
&\geq \sum_k [\mathbf{1}_{q_k \geq 0} \cdot \lambda_{min} \cdot q_k + \mathbf{1}_{q_k < 0} \cdot \lambda_{max} \cdot q_k] \\
&= \sum_k [\mathbf{1}_{q_k \geq 0} \cdot \lambda_l \cdot q_k + \mathbf{1}_{q_k < 0} \cdot \lambda_l \cdot q_k] \\
&= \lambda_l \cdot J_{avg}(\theta_0)^\top \cdot J_j(\theta_0) \\
&= \lambda_l \cdot a_j,
\end{aligned}
$$

which is the lower bound in Lemma 4 $\qquad\square$

### C.3 Choosing the scaling factor $C$

By Lemma 4, the influence function estimate $a_j^{\text{IF}}$ is scaled by the GUARD proxy attribution $a_j^{GUARD}$, satisfying the bound:

$$
\lambda_l \cdot a_j^{GUARD} \leq a_j^{\text{IF}} \leq \lambda_u \cdot a_j^{GUARD}.
$$

Here, the upper and lower scaling factors are given by

$$
\lambda_u := \frac{\lambda_{\max} q^+ + \lambda_{\min} q^-}{q^+ + q^-}, \quad \lambda_l := \frac{\lambda_{\min} q^+ + \lambda_{\max} q^-}{q^+ + q^-},
$$

where

$$
q^+ := \sum_k \mathbf{1}_{q_k \geq 0} \cdot q_k, \quad q^- := \sum_k \mathbf{1}_{q_k < 0} \cdot q_k.
$$

The multipliers $\lambda_u$ and $\lambda_l$ are data-dependent, varying with each specific sample $(x_j, y_j)$. To emphasize this dependence, we denote the sample-specific bounds as $\lambda_l^j$ and $\lambda_u^j$. Consequently, for each $j$, there exists a constant $C_j \in [\lambda_l^j, \lambda_u^j]$ such that

$$
\hat{\Delta}_L^j = C_j \cdot a_j^{GUARD}.
$$

Approximating $\hat{\Delta}_L^j$ by a globally optimized scaling factor $C$, we obtain the following approximation error

$$
\text{err} := \frac{1}{n_f} \sum_{j \in I_f} \|\hat{\Delta}_L^j - C \cdot a_j^{GUARD}\|_2^2 = \frac{1}{n_f} \sum_{j \in I_f} (C_j - C)^2 \cdot [a_j^{GUARD}]^2.
$$

Minimizing this error with respect to $C$, the optimal scaling factor is given by

$$
C^* = \overline{C}_{GUARD},
$$

where $\overline{C}_{GUARD} := \frac{\sum_{j \in I_f} C_j \cdot [a_j^{GUARD}]^2}{\sum_{j \in I_f} [a_j^{GUARD}]^2}$ is the average of $\{C_j\}$ weighted by $[a_j^{GUARD}]^2$.

# D PROOFS OF THEORETICAL GUARANTEES

*Lemma* 5 (Restatement of Lemma 1). *Under Assumption 1, the loss on the retain set under GUARD is lower than that under GA by:*

$$L_{\mathcal{D}_r}^{GA} - L_{\mathcal{D}_r}^{GUARD} = \frac{\eta}{(1 - \delta_\kappa)\tau} \cdot \sigma_\kappa^2 + \mathcal{O}(\delta^2),$$

*where $\eta$ is the unlearning rate.*

*Proof.* Recall that the losses of different models on the retain set are defined as

$$L_{\mathcal{D}_r}^0 = \frac{1}{n_r} \sum_{i \in I_r} l(x_i, y_i; \theta^0),$$

$$L_{\mathcal{D}_r}^{GA} = \frac{1}{n_r} \sum_{i \in I_r} l(x_i, y_i; \theta^{GA}),$$

$$L_{\mathcal{D}_r}^{GUARD} = \frac{1}{n_r} \sum_{i \in I_r} l(x_i, y_i; \theta^{GUARD})$$

Correspondingly, the model update rules given by GA and by GUARD are respectively

$$\theta^{GA} = \theta^0 + \eta \cdot \frac{1}{n_f} \sum_{i \in I_f} g_i$$

$$\theta^{GUARD} = \theta^0 + \eta \cdot \frac{1}{n_f} \sum_{i \in I_f} [\omega_i^{GUARD} \cdot g_i],$$

where

$$\omega_i^{GUARD} := n_f \cdot \frac{e^{-a_i^{GUARD}/\tau}}{\sum_{j \in I_f} e^{-a_j^{GUARD}/\tau}}, \quad i \in I_f. \tag{9}$$

Based on the loss definitions and model update rules, we get the approximations of the increase in the retain loss compared with the model before unlearning. Specifically, the increase in the loss on the retain set induced by GA is given by

$$
\begin{aligned}
L_{\mathcal{D}_r}^{GA} - L_{\mathcal{D}_r}^0 &= \frac{1}{n_r} \sum_{i \in I_r} g_i^\top \cdot [\theta^{GA} - \theta^0] + \mathcal{O}(\delta_\theta^2) \\
&= \frac{1}{n_r} \sum_{i \in I_r} g_i^\top \cdot [\eta \cdot \frac{1}{n_f} \sum_{j \in I_f} g_j] + \mathcal{O}(\delta_\theta^2) \\
&= \eta \cdot \langle \overline{g}_f, \overline{g}_r \rangle + \mathcal{O}(\delta_\theta^2) \\
&= \eta \cdot \kappa + \mathcal{O}(\delta_\theta^2),
\end{aligned}
\tag{10}
$$

and the increase in the loss on the retain set induced by GUARD is given by

$$
\begin{aligned}
L_{\mathcal{D}_r}^{GUARD} - L_{\mathcal{D}_r}^0 &= \frac{1}{n_r} \sum_{i \in I_r} g_i^\top \cdot [\theta^{GUARD} - \theta^0] + \mathcal{O}(\delta_\theta^2) \\
&= \frac{1}{n_r} \sum_{i \in I_r} g_i^\top \cdot [\eta \cdot \frac{1}{n_f} \sum_{i \in I_f} [\omega_i^{GUARD} \cdot g_i]] + \mathcal{O}(\delta_\theta^2) \\
&= \eta \cdot \langle \overline{g}_r, \overline{g}_f' \rangle + \mathcal{O}(\delta_\theta^2),
\end{aligned}
\tag{11}
$$

where $\overline{g}'_f := \frac{1}{n_f} \sum_{i \in I_f} \left[ \omega_i^{\text{GUARD}} \cdot g_i \right]$. Let $\kappa' := \langle \overline{g}_r, \overline{g}'_f \rangle$. It holds that

$$
\kappa' = \frac{1}{n_r} \sum_{i \in I_r} g_i^\top \cdot \frac{1}{n_f} \sum_{j \in I_f} \omega_j^{\text{GUARD}} \cdot g_j
$$

$$
= \frac{1}{n_r \cdot n_f} \sum_{i \in I_r} g_i^\top \cdot \sum_{j \in I_f} \left[ n_f \cdot \frac{e^{-a_j^{\text{GUARD}}/\tau}}{\sum_{k \in I_f} e^{-a_k^{\text{GUARD}}/\tau}} \right] \cdot g_j
$$

$$
= \frac{1}{n_r \cdot n_f} \sum_{i \in I_r} g_i^\top \cdot \sum_{j \in I_f} \left[ n_f \cdot \frac{1 - a_j^{\text{GUARD}}/\tau}{\sum_{k \in I_f} [1 - a_k^{\text{GUARD}}/\tau]} \right] \cdot g_j + \mathcal{O}(\delta_\kappa^2)
$$

$$
= \frac{1}{n_r \cdot n_f} \sum_{i \in I_r} g_i^\top \cdot \sum_{j \in I_f} \left[ \frac{1 - \overline{g}_r^\top \cdot g_j/\tau}{1 - \overline{g}_r^\top \cdot \overline{g}_f/\tau} \right] \cdot g_j + \mathcal{O}(\delta_\kappa^2)
$$

$$
= \frac{1}{n_r \cdot n_f \cdot [1 - \frac{\kappa}{\tau}]} \sum_{i \in I_r} g_i^\top \sum_{j \in I_f} [g_j - \overline{g}_r^\top g_j/\tau \cdot g_j] + \mathcal{O}(\delta_\kappa^2)
$$

$$
= \frac{1}{n_r \cdot n_f \cdot [\tau - \kappa]} \left[ n_r \cdot n_f \cdot \kappa \cdot \tau - n_r \cdot \overline{g}_r^\top \cdot \sum_{j \in I_f} [\overline{g}_r^\top g_j \cdot g_j] \right] + \mathcal{O}(\delta_\kappa^2)
$$

$$
= \frac{\tau \cdot \kappa}{\tau - \kappa} - \frac{1}{n_f \cdot [\tau - \kappa]} \cdot \sum_{j \in I_f} [\overline{g}_r^\top g_j]^2 + \mathcal{O}(\delta_\kappa^2)
$$

With a slight abuse of notation, define the alignment between averaged gradients of the Retain Set and datawise gradients of the Forget Set as $\kappa_j := \langle \overline{g}_r, g_j \rangle$, $j \in I_f$. Define its variance as $\sigma_\kappa^2 := \frac{1}{n_f} \sum_{j \in I_f} [\kappa_j]^2 - [\frac{1}{n_f} \sum_{j \in I_f} \kappa_j]^2$, equivalent to $\sum_{j \in I_f} \kappa_j^2 = n_f \cdot (\sigma_\kappa^2 + \kappa^2)$. Therefore,

$$
\kappa' = \frac{\tau \cdot \kappa}{\tau - \kappa} - \frac{1}{\tau - \kappa} \cdot [\sigma_\kappa^2 + \kappa^2] + \mathcal{O}(\delta_\kappa^2) \tag{12}
$$

Substitute Eq. 12 and Eq. 11 into Eq. 10 and get

$$
L_{\mathcal{D}_r}^{\text{GA}} - L_{\mathcal{D}_r}^{\text{GUARD}} = (1 - \delta_\kappa)^{-1} \cdot \eta/\tau \cdot \sigma_\kappa^2 + \mathcal{O}(\delta^2).
$$

$\qquad\qquad\qquad\qquad\qquad\qquad\qquad\qquad\qquad\qquad\qquad\qquad\qquad\qquad\qquad\qquad\qquad\qquad\qquad\quad\square$

*Lemma* 6 (Restatement of Lemma 2). *Under Assumption 1, the difference in the forget loss between GUARD and GA is bounded as*

$$
\left| L_{\mathcal{D}_f}^{GA} - L_{\mathcal{D}_f}^{GUARD} \right| = \delta_\kappa \eta \cdot \|\overline{g}_f\|_2^2 + \mathcal{O}(\delta^2).
$$

*Proof.* Recall that the losses of different models on the forget set are defined as

$$
L_{\mathcal{D}_f}^0 = \frac{1}{n_f} \sum_{i \in I_f} l(x_i, y_i; \theta^0),
$$

$$
L_{\mathcal{D}_f}^{\text{GA}} = \frac{1}{n_f} \sum_{i \in I_f} l(x_i, y_i; \theta^{GA}),
$$

$$
L_{\mathcal{D}_f}^{\text{GUARD}} = \frac{1}{n_f} \sum_{i \in I_f} l(x_i, y_i; \theta^{\text{GUARD}}).
$$

Correspondingly, the model update rules given by GA and by GUARD are respectively

$$
\theta^{\text{GA}} = \theta^0 + \eta \cdot \frac{1}{n_f} \sum_{i \in I_f} g_i
$$

$$
\theta^{\text{GUARD}} = \theta^0 + \eta \cdot \frac{1}{n_f} \sum_{i \in I_f} [\omega_i^{\text{GUARD}} \cdot g_i]
$$

Suppose that the second and higher orders of the norm of change in model weights can be omitted in the theoretical analysis. Then, based on loss definitions and model update rules, we get the approximations of the increase in the forget loss compared with the model before unlearning. Specifically, the increase in the loss on the forget set induced by GA is given by

$$L_{\mathcal{D}_f}^{\text{GA}} - L_{\mathcal{D}_f}^0 = \frac{1}{n_f} \sum_{i \in I_f} g_i^\top \cdot [\theta^{\text{GA}} - \theta^0] + \mathcal{O}(\delta_\theta^2)$$

$$= \frac{1}{n_f} \sum_{i \in I_f} g_i^\top \cdot [\eta \cdot \frac{1}{n_f} \sum_{j \in I_f} g_j] + \mathcal{O}(\delta_\theta^2)$$

$$= \eta \cdot \langle \overline{g}_f, \overline{g}_f \rangle + \mathcal{O}(\delta_\theta^2), \tag{13}$$

and the increase in the loss on the Forget Set induced by GUARD is given by

$$L_{\mathcal{D}_f}^{\text{GUARD}} - L_{\mathcal{D}_f}^0 = \frac{1}{n_f} \sum_{i \in I_f} g_i^\top \cdot [\theta^{\text{GUARD}} - \theta^0] + \mathcal{O}(\delta_\theta^2)$$

$$= \frac{1}{n_f} \sum_{i \in I_f} g_i^\top \cdot \left[ \eta \cdot \frac{1}{n_f} \sum_{j \in I_f} [\omega_j^{\text{GUARD}} \cdot g_j] \right] + \mathcal{O}(\delta_\theta^2)$$

$$= \eta \cdot \langle \overline{g}_f', \overline{g}_f \rangle + \mathcal{O}(\delta_\theta^2), \tag{14}$$

where $\overline{g}_f'$ is the weighted average gradient over forget set, $\overline{g}_f' := \frac{1}{n_f} \sum_{i \in I_f} [\omega_i^{\text{GUARD}} \cdot g_i]$. Also,

$$\langle \overline{g}_f, \overline{g}_f' \rangle = \left[ \frac{1}{n_f} \sum_{i \in I_f} g_i \right]^\top \cdot \left[ \frac{1}{n_f} \sum_{j \in I_f} [\omega_j^{\text{GUARD}} \cdot g_j] \right]$$

$$= \frac{1}{n_f^2} \sum_{i,j \in I_f} \left[ n_f \cdot \frac{e^{-a_j^{\text{GUARD}}/\tau}}{\sum_{k \in I_f} e^{-a_k^{\text{GUARD}}/\tau}} \cdot g_i^\top \cdot g_j \right]$$

$$= \frac{1}{n_f} \sum_{i,j \in I_f} \left[ \frac{1 - \overline{g}_r^\top \cdot g_j/\tau}{\sum_{k \in I_f} [1 - \overline{g}_r^\top \cdot g_k/\tau]} \cdot g_i^\top \cdot g_j \right] + \mathcal{O}(\delta_\kappa^2)$$

$$= \frac{1}{n_f^2} \cdot \frac{\tau}{\tau - \kappa} \sum_{i,j \in I_f} \left[ [1 - \overline{g}_r^\top \cdot g_j/\tau] \cdot g_i^\top \cdot g_j \right] + \mathcal{O}(\delta_\kappa^2)$$

$$= \frac{\tau}{n_f(\tau - \kappa)} \left[ n_f \|\overline{g}_f\|_2^2 - \frac{1}{\tau} \sum_{j \in I_f} [\overline{g}_r^\top \cdot g_j \cdot g_j^\top \cdot \overline{g}_f] \right] + \mathcal{O}(\delta_\kappa^2). \tag{15}$$

Let $d$ be the dimensionality of the gradient vectors. Then we can write

$$\sum_{j \in I_f} \overline{g}_r^\top g_j \cdot g_j^\top \overline{g}_f = \overline{g}_r^\top \left( \sum_{j \in I_f} g_j g_j^\top \right) \overline{g}_f$$

$$= n_f \cdot \overline{g}_r^\top \Sigma_f \overline{g}_f$$

$$\approx n_f \cdot \lambda \cdot \overline{g}_r^\top \overline{g}_f$$

$$= n_f \cdot \lambda \cdot \kappa. \tag{16}$$

Let $\text{Cov}_f$ be the sample covariance matrix of $g_j$, $j \in I_f$, defined as $\text{Cov}_f := \frac{1}{n_f} \sum_{j \in I_f} (g_j - \overline{g}_f)(g_j - \overline{g}_f)^\top$. Then we have

$$
\begin{aligned}
\lambda &:= \frac{1}{d} \text{Tr}(\Sigma_f) \\
&= \frac{1}{d} \text{Tr}(\overline{g}_f \overline{g}_f^\top + \text{Cov}_f) \\
&= \frac{1}{d} \left( \|\overline{g}_f\|_2^2 + \text{Tr}(\text{Cov}_f) \right) \\
&= \frac{1}{d} \left( \|\overline{g}_f\|_2^2 + \text{Var}_g \right),
\end{aligned} \tag{17}
$$

where $\text{Var}_f$ is the total variance of the gradients defined as $\text{Var}_f := \sum_{k=1}^{d} \text{Var}\left[(g_i)_k\right]$, where $(g_i)_k$ denotes the $k$-th coordinate of the gradient vector $g_i$.

Substituting Eq. 16 and Eq. 17 into Eq. 15, we get:

$$
\begin{aligned}
\langle \overline{g}_f', \overline{g}_f \rangle &= \frac{\tau}{\tau - \kappa} \|\overline{g}_f\|_2^2 - \frac{\kappa}{(\tau - \kappa) \cdot d} \left( \|\overline{g}_f\|_2^2 + \text{Var}_g \right) + \mathcal{O}(\delta_\kappa^2) \\
&= (1 - \kappa/\tau)^{-1} \|\overline{g}_f\|_2^2 - \frac{\kappa/\tau}{1 - \kappa/\tau} \cdot d^{-1} \cdot \left( \|\overline{g}_f\|_2^2 + \text{Var}_g \right) + \mathcal{O}(\delta_\kappa^2) \\
&= (1 + \kappa/\tau) \|\overline{g}_f\|_2^2 - \kappa/\tau \cdot (1 + \kappa/\tau) \cdot d^{-1} \cdot \left( \|\overline{g}_f\|_2^2 + \text{Var}_g \right) + \mathcal{O}(\delta_\kappa^2) \\
&= (1 + \delta_\kappa) \|\overline{g}_f\|_2^2 + \mathcal{O}(\delta_\kappa^2).
\end{aligned} \tag{18}
$$

Substituting Eq. 18 into Eq. 14, we get

$$
L_{\mathcal{D}_f}^{\text{GUARD}} - L_{\mathcal{D}_f}^0 = \eta \cdot (1 + \delta_\kappa) \|\overline{g}_f\|_2^2 + \mathcal{O}(\delta^2) \tag{19}
$$

Combining Eq. 19 with 13, we get

$$
\left| L_{\mathcal{D}_f}^{\text{GA}} - L_{\mathcal{D}_f}^{\text{GUARD}} \right| = \eta \cdot \delta_\kappa \cdot \|\overline{g}_f\|_2^2 + \mathcal{O}(\delta^2). 
$$

$\square$

*Theorem* 7 (Restatement of Theorem 3). *Under Assumption 1, GUARD reduces the Sacrifice Rate relative to GA as:*

$$
\rho^{GA} - \rho^{GUARD} = \frac{\kappa^2 + \sigma_\kappa^2}{\tau \cdot \|\overline{g}_f\|_2^2} + \mathcal{O}(\delta^2).
$$

*Proof.* Rewrite Eq. 12 as

$$
\begin{aligned}
\kappa' &= \frac{\tau \cdot \kappa}{\tau - \kappa} - \frac{1}{\tau - \kappa} \cdot [\sigma_\kappa^2 + \kappa^2] + \mathcal{O}(\delta_\kappa^2) \\
&= \frac{\kappa - \sigma_\kappa^2/\tau - \kappa^2/\tau}{1 - \kappa/\tau} + \mathcal{O}(\delta_\kappa^2) \\
&= (\kappa - \sigma_\kappa^2/\tau - \kappa/\tau \cdot \kappa) \cdot (1 + \kappa/\tau) + \mathcal{O}(\delta_\kappa^2) \\
&= \kappa - \sigma_\kappa^2/\tau - \sigma_\kappa^2 \cdot \kappa/\tau^2 + \mathcal{O}(\delta^2).
\end{aligned} \tag{20}
$$

Substituting Eq. 20 into Eq. 11, we have

$$
L_{\mathcal{D}_r}^{\text{GUARD}} - L_{\mathcal{D}_r}^0 = \eta \cdot (\kappa - \sigma_\kappa^2/\tau - \sigma_\kappa^2 \cdot \kappa/\tau^2) + \mathcal{O}(\delta_\kappa^2). \tag{21}
$$

Combining Eq. 10 and Eq. 13 we get

$$
\rho^{\text{GA}} = \frac{L_{\mathcal{D}_r}^{\text{GA}} - L_{\mathcal{D}_r}^0}{L_{\mathcal{D}_f}^{\text{GA}} - L_{\mathcal{D}_f}^0} = \frac{\eta \kappa}{\eta \|\overline{g}_f\|_2^2} + \mathcal{O}(\delta^2) = \|\overline{g}_f\|_2^{-2} \cdot \kappa + \mathcal{O}(\delta^2). \tag{22}
$$

Combining Eq. 21 and Eq. 19 we get

$$
\begin{aligned}
\rho^{\text{GUARD}} &= \frac{L_{\mathcal{D}_r}^{\text{GUARD}} - L_{\mathcal{D}_r}^0}{L_{\mathcal{D}_f}^{\text{GUARD}} - L_{\mathcal{D}_f}^0} \\
&= \frac{\eta \cdot (\kappa - \sigma_\kappa^2/\tau - \sigma_\kappa^2 \cdot \kappa/\tau^2)}{\eta \cdot (1 + \kappa/\tau)\|\bar{g}_f\|_2^2} + \mathcal{O}(\delta^2) \\
&= \|\bar{g}_f\|_2^{-2} \cdot (\kappa - \sigma_\kappa^2/\tau - \sigma_\kappa^2/\tau \cdot \kappa/\tau)(1 - \kappa/\tau) + \mathcal{O}(\delta^2) \\
&= \|\bar{g}_f\|_2^{-1} \cdot (\kappa - \sigma_\kappa^2/\tau - \kappa^2/\tau) + \mathcal{O}(\delta^2)
\end{aligned}
\tag{23}
$$

Combining Eq. 22 and Eq. 23 we get

$$
\rho^{\text{GA}} - \rho^{\text{GUARD}} = \frac{\kappa^2 + \sigma_\kappa^2}{\tau \cdot \|\bar{g}_f\|_2^2} + \mathcal{O}(\delta^2).
$$

□

### D.1 ANALYSIS OF KEY FACTORS FOR THEORETICAL GUARANTEES

1. **Unlearning rate $\eta$.** An increased unlearning rate $\eta$ improves retention on the Retain Set $\mathcal{D}_r$ by amplifying gradient alignment (Lemma 1). However, it may slightly reduce unlearning effectiveness by overcorrecting the model on $\mathcal{D}_f$ (Lemma 2). Overall, a higher $\eta$ leads to a greater reduction in the Sacrifice Rate (Theorem 3).

2. **Temperature $\tau$.** Lowering the temperature induces a sharper attribution distribution across data points. This has three effects: (1) it can enhance retention by emphasizing high-variance alignment directions (Lemma 1); (2) it may weaken unlearning effectiveness due to increased entanglement between $\mathcal{D}_f$ and $\mathcal{D}_r$ (Lemma 2); and (3) it can result in a larger reduction in the Sacrifice Rate, especially when $\delta_\kappa = \kappa/\tau \ll 1$ (Theorem 3).

3. **Knowledge alignment $\kappa = \langle \bar{g}_f, \bar{g}_r \rangle$.** A smaller value of $\kappa$ indicates weaker coupling between the forget and retain gradients. This leads to improved retention (due to reduced interference), more effective unlearning (as $\mathcal{D}_f$ updates minimally affect $\mathcal{D}_r$), and a greater reduction in the Sacrifice Rate.

4. **Variance of datawise alignment $\sigma_\kappa^2$.** A higher variance $\sigma_\kappa^2$ implies greater heterogeneity in the alignment between individual examples in $\mathcal{D}_f$ and $\mathcal{D}_r$. This allows GUARD to better disentangle retention and forgetting. As a result, retention improves, forget effectiveness remains largely stable, and the Sacrifice Rate reduction is more pronounced.

5. **Norm of the average forget gradient $\|\bar{g}_f\|$.** Larger values of $\|\bar{g}_f\|$ have minimal effect on retention. However, they hinder unlearning by making the forget gradients more dominant, thereby resisting suppression (Lemma 2). This also leads to a smaller reduction in the Sacrifice Rate due to their influence on the denominator of the alignment term (Theorem 3).

## E ALGORITHMS

### E.1 UPDATE RULES OF BASELINE METHODS

We list the baseline unlearning algorithms provided in TOFU benchmark Maini et al. (2024b), including Gradient Ascent (GA), Gradient Difference (GD), KL Minimization (KM), and Preference Optimization (PO).

**Gradient Ascent (GA)**. The Gradient Ascent method aims to reduce the model's confidence in its initial predictions on the forget set. Specifically, for each instance in $D_f$, the goal is to maximize the training loss, encouraging the model to diverge from its prior learned outputs, and the overall loss objective is defined as:

$$
L_{GA} = -L_{D_f}(\theta)
\tag{24}
$$

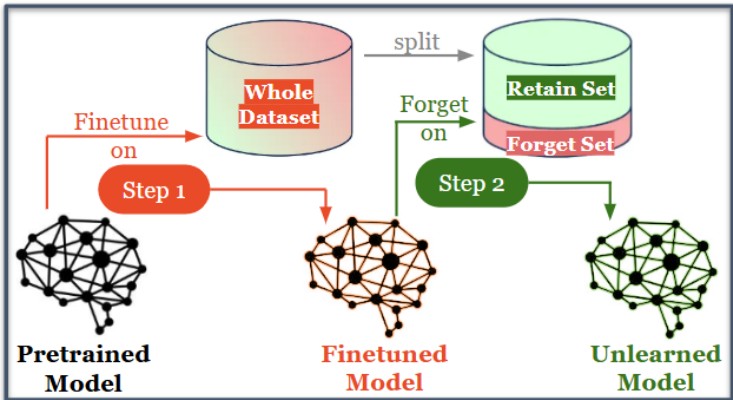

Figure 2: Illustration of the standard unlearning pipeline for LLMs, which is applying an unlearning algorithm to remove the influence of a specific subset of data (Forget Set).

**Gradient Difference (GD)** Lu et al. (2022). The Gradient Difference method builds upon the gradient ascent approach by not only increasing loss on the forget set $D_f$ but also preserving model performance on the retain set $D_r$. The objective function for this approach is formulated as:

$$L_{\text{GD}} = -L_{D_F}(\theta) + L_{D_r}(\theta). \tag{25}$$

To balance computational cost, for each sample processed from $D_f$, a corresponding example from $D_r$, ensuring the model remains within feasible resource constraints.

**KL Divergence Minimization (KM)**. The KL Minimization technique aims to regulate the model's unlearning process by ensuring that predictions on the retain set $D_r$ remain consistent with those of the original model while still reducing accuracy on $D_f$. Let $M$ be the model function, where $M(\cdot)$ produces a probability distribution over possible token predictions. The training objective is expressed as:

$$L_{\text{KM}} = -L_{D_f}(\theta) + \frac{1}{|D_r|} \sum_{s \in D_r} \frac{1}{|s|} \sum_{i=2}^{|s|} \text{KL}\big(M_0(s_{<i}) \parallel M_{\text{u}}(s_{<i})\big). \tag{26}$$

Here, $M_0$ and $M_{\text{u}}$ refer to the model before and after unlearning, respectively. Due to resource constraints, instances from $D_r$ are sampled randomly, while the entire forget set is used.

**Preference-Based Optimization**. Inspired by Direct Preference Optimization (DPO) (Rafailov et al., 2023), this method substitutes responses in $D_f$ with alternative neutral responses such as "I do not have that information". The loss function is as follows:

$$L_{\text{PO}} = L_{D_r}(\theta) + L_{D_f^{\text{idk}}}(\theta). \tag{27}$$

This ensures that while the model updates its responses for $D_f$, its overall linguistic abilities and predictive accuracy on $D_r$ remain stable.

### E.2 GUARD OBJECTIVE FOR FRAMEWORKS OTHER THAN GA

Given the forget dataset $D_f$ and a model parameterized by $\theta$, with slight abuse of notations, the GUARD loss for additional frameworks are defined as follows:

- GUARD objective under Gradient Difference.

$$L_{GD}^{GUARD}(\theta) := -L_{D_F}^{GUARD}(\theta) + L_{D_r}(\theta)$$

- GUARD objective under KL Minimization.

$$L_{\text{KM}}^{GUARD} = -L_{D_f}^{GUARD}(\theta) + \frac{1}{|D_r|} \sum_{s \in D_r} \frac{1}{|s|} \sum_{i=2}^{|s|} \text{KL}\big(M_0(s_{<i}) \parallel M_{\text{u}}(s_{<i})\big). \tag{28}$$

- GUARD objective under Preference Optimization.

$$L_{\text{PO}}^{GUARD} = L_{D_r}(\theta) + L_{D_f^{\text{idk}}}^{GUARD}(\theta).$$

(29)

- GUARD objective under Negative Preference Optimization.

$$L_{\text{NPO}}^{GUARD} = L_{D_f,\text{NPO}}^{GUARD}(\theta) := \frac{1}{n_f} \sum_{i \in I_f} \omega_i^{GUARD} \cdot \frac{2}{\beta} \log \left( 1 + \left( \frac{\pi_\theta(y_i|x_i)}{\pi_{\text{ref}}(y_i|x_i)} \right)^\beta \right).$$

(30)

### E.3 GUARD ALGORITHMS FOR ADDITIONAL BASELINES

---

**Algorithm 2** GUARD: Guided Unlearning and Retention via Data Attribution

---

1: **Input**: Fine-tuned model weights $\theta_0$, Unlearning rate $\eta$, Baseline method $B$.
2: Compute attribution scores $a_i$ for all $i \in I_f$ ▷ Estimate data attribution for forget set
3: Compute GUARD multipliers $\omega_i$ for all $i \in I_f$ using Eq. 3 ▷ Assign unlearning weights
4: **if** $B$ is GD **then** ▷ For Gradient Difference
5:     Compute unlearned model parameters $\theta_{\text{GD}}^{\text{GUARD}}$ using Eq. 25 ▷ GUARD Unlearning
6: **else**
7:     **if** $B$ is KM **then** ▷ For KL Minimization
8:         Compute unlearned model parameters $\theta_{\text{KM}}^{\text{GUARD}}$ using Eq. 26 ▷ GUARD Unlearning
9:     **else**
10:         **if** $B$ is PO **then** ▷ For Preference Optimization
11:             Compute unlearned model parameters $\theta_{\text{PO}}^{\text{GUARD}}$ using Eq. 29 ▷ GUARD Unlearning
12:         **else if** $B$ is NPO **then** ▷ For Negative Preference Optimization
13:             Compute unlearned model parameters $\theta_{\text{NPO}}^{\text{GUARD}}$ using Eq. 30 ▷ GUARD Unlearning
14:         **end if**
15:     **end if**
16: **end if**
17: **Output**: Unlearned model weights $\theta_B^{GUARD}$.

---

## F ADDITIONAL RESULTS

### F.1 ADDITIONAL SETTINGS

**Configuration and Hyperparameters.** All experiments were evaluated on equal-cost hardware, Ubuntu 22.04 LTS system with 2 Intel(R) Xeon(R) Gold 6138 CPUs with 20 cores, 4 NVIDIA Tesla V100-SXM2 GPUs with 32 GB memory, and 384 GB RAM. All experiments use 40 CPU cores and 4 GPUs. Unless otherwise specified, the following configurations are used:

TOFU benchmark: During the fine-tuning phase, we train for 5 epochs using a learning rate of $10^{-5}$ and a weight decay of 0.01. The batch size is set to 4 for Phi-1.5B and 1 for LLaMA-2-7B. During the data attribution phase, we use a threshold of $\tau = 0.03$ unless stated otherwise. In the forgetting phase, by default, each experiment use 1 training epoch, a batch size of 1, a learning rate of $10^{-5}$, and a weight decay of 0.01.

MUSE benchmark: MUSE provides pre-finetuned large models, which can be directly downloaded from its Huggingface repository. During the data attribution phase, we use a threshold of $\tau = 0.03$ unless stated otherwise. In the forgetting phase, by default, each experiment use exactly the same setting compare to MUSE benchmark's instruction.

### F.2 RESULTS FOR LLAMA-2-7B ON THE TOFU DATASET

We present the results in Table 4, 5.

### F.3 RESULTS FOR PHI-1.5B ON THE TOFU DATASET

We present the results of Phi-1.5B in Tables 6, 7 and 8.

Table 4: Unlearning experiments with LLaMA-2-7B on 1% of all data points in TOFU. GUARD consistently reduces the Sacrifice Rates across all baselines and evaluation metrics, especially on the Retain Set. It can also lead to negative Sacrifice Rates (i.e., performance gains) on generalization sets, suggesting enhanced generalization via reduction of overfitting on the forget set.

| Methods | Retain Set($\downarrow$) | | | Real Author Set($\downarrow$) | | | Real World Set($\downarrow$) | | |
|---|---|---|---|---|---|---|---|---|---|
| | $\rho_r$ | $\rho_p$ | $\rho_t$ | $\rho_r$ | $\rho_p$ | $\rho_t$ | $\rho_r$ | $\rho_p$ | $\rho_t$ |
| GA | 87.87 | 91.25 | 289.06 | 58.15 | 73.91 | 172.38 | 31.54 | 64.55 | 119.48 |
| GA + GUARD | **49.82** | **55.32** | **124.73** | **39.59** | **52.36** | **90.89** | **17.68** | **-18.82** | **-5.93** |
| KM | 82.45 | 79.01 | 134.99 | 41.42 | 18.10 | 129.15 | 5.96 | 19.09 | 59.17 |
| KM + GUARD | **43.42** | **45.08** | **24.69** | **27.16** | **0.18** | **55.38** | **-2.94** | **-7.08** | **-1.13** |
| PO | 24.59 | 47.38 | 94.13 | 16.39 | 14.87 | 61.95 | -15.34 | 5.84 | -21.89 |
| PO + GUARD | **9.36** | **31.52** | **15.99** | **6.42** | **4.55** | **51.36** | **-17.40** | **-2.22** | **-29.21** |
| GD | 11.97 | 12.15 | 91.65 | 2.63 | 4.49 | -15.86 | -14.29 | -17.52 | -29.21 |
| GD + GUARD | **1.64** | **8.98** | **15.51** | **-1.53** | **-3.78** | **-18.04** | **-17.91** | **-18.59** | **-32.02** |

Table 5: Unlearning experiments with LLaMA-2-7B on 5% of all data points in TOFU. GUARD consistently reduces the Sacrifice Rates across all evaluation metrics and baselines, with particularly large improvements over baselines that exhibiting weak retention (the most substantial gains are observed against GA, which has the highest Sacrifice Rates).

| Methods | Retain Set($\downarrow$) | | | Real Author Set($\downarrow$) | | | Real World Set($\downarrow$) | | |
|---|---|---|---|---|---|---|---|---|---|
| | $\rho_r$ | $\rho_p$ | $\rho_t$ | $\rho_r$ | $\rho_p$ | $\rho_t$ | $\rho_r$ | $\rho_p$ | $\rho_t$ |
| GA | 90.57 | 118.59 | 401.01 | 80.77 | 88.02 | 313.25 | 64.76 | 37.84 | 216.97 |
| GA + GUARD | **64.31** | **87.93** | **237.29** | **61.73** | **65.49** | **236.95** | **36.65** | **5.32** | **165.84** |
| KM | 86.15 | 112.50 | 337.22 | 78.74 | 42.56 | 180.75 | 20.50 | 20.25 | 98.92 |
| KM + GUARD | **60.60** | **86.79** | **201.89** | **58.10** | **25.17** | **134.08** | **-5.24** | **-11.51** | **66.66** |
| PO | 47.96 | 104.14 | 200.59 | 42.45 | 25.22 | 152.90 | 6.24 | -9.37 | 19.02 |
| PO + GUARD | **25.24** | **82.39** | **118.83** | **25.92** | **13.25** | **97.87** | **-4.73** | **-12.18** | **-11.29** |
| GD | 40.97 | 79.47 | 114.25 | 21.40 | -14.44 | -17.61 | -21.13 | -11.53 | -11.50 |
| GD + GUARD | **18.71** | **59.30** | **86.16** | **10.49** | **-18.52** | **-23.77** | **-27.13** | **-13.95** | **-23.74** |

**Effectiveness of GUARD.**  Across all datasets, methods, and splits, GUARD consistently reduces the sacrifice rate across key metrics ($\rho_r$, $\rho_p$, and $\rho_t$), demonstrating its ability to mitigate unintended knowledge degradation while effectively removing targeted data. Specifically:

- *Forgetting 1% of data:* The largest improvement occurs on the Retain Set, where GUARD reduces $\rho_r$ by 9.77% (compared to GA), $\rho_p$ by 16.33% (compared to PO), and $\rho_t$ by 16.35% (compared to KM).

- *Forgetting 5% of data:* The largest improvement is again on the Retain Set, where GUARD lowers $\rho_r$ by 23% (compared to KM), $\rho_p$ by 26% (compared to GA), and $\rho_t$ by 18.02% (compared to GA).

- *Forgetting 10% of data:* The most significant improvement is observed on the Retain Set, with reductions of $\rho_r$ by 23.7% (compared to PO), $\rho_p$ by 50.44% (compared to PO), and $\rho_t$ by 54.75% (compared to GA).

**Impact of Forgetting Percentage.**  Larger unlearning proportions generally lead to higher sacrifice rates, aligning with the intuition that removing a greater volume of data negatively affects knowledge retention. However, as the forgetting proportion increases, GUARD also achieves more substantial reductions in sacrifice rates across all evaluation metrics, baselines, and datasets. For instance, in the case of $\rho_r$, the largest reductions by GUARD for forgetting 1%, 5%, and 10% of the total datapoints are 16.35%, 18.02%, and 54.75%, respectively.

Table 6: Sacrifice rate (%) of different unlearning methods when unlearning Phi-1.5B on 1% of TOFU data. Across all datasets (Retain Set, Real Author Set, and Real World Set), GUARD consistently reduces the sacrifice rate across various methods (GA, GD, KM, PO). The largest improvement is observed on the Retain Set, where GUARD lowers $\rho_r$ by 9.77% over GA, $\rho_p$ by 16.33% over PO, and $\rho_t$ by 16.35% over KM. Negative values in the Real Author and Real World Sets suggest that GUARD enhances generalization, likely by reducing overfitting on the forget set.

| Methods | Retain Set | | | Real Author Set | | | Real World Set | | |
|---|---|---|---|---|---|---|---|---|---|
| | $\rho_r$ | $\rho_p$ | $\rho_t$ | $\rho_r$ | $\rho_p$ | $\rho_t$ | $\rho_r$ | $\rho_p$ | $\rho_t$ |
| GA | 17.99 | 33.55 | 87.17 | -2.77 | 1.65 | 23.29 | 3.90 | -2.26 | 35.13 |
| GA+GUARD | **8.22** | **30.16** | **69.12** | **-9.18** | **1.21** | **9.78** | **-3.44** | **-3.03** | **28.27** |
| GD | 9.16 | 15.95 | -0.50 | -0.83 | -4.37 | -0.50 | 0.81 | 3.63 | -1.57 |
| GD+GUARD | **8.71** | **13.11** | **-0.85** | **-3.44** | **-6.55** | **0.23** | **2.74** | **-2.47** | **-4.66** |
| KM | 11.75 | 31.28 | 53.78 | -2.46 | 0.59 | 8.19 | 0.64 | -2.83 | -31.68 |
| KM+GUARD | **5.97** | **27.43** | **37.43** | **-8.39** | **-0.70** | **-0.38** | **-1.35** | **-3.57** | **-34.63** |
| PO | 11.90 | 69.16 | 21.74 | -3.60 | 12.84 | 9.47 | -3.85 | -1.86 | -2.40 |
| PO+GUARD | **9.04** | **52.83** | **5.68** | **-7.50** | **3.95** | **3.67** | **-7.09** | **-7.35** | **-10.50** |

Table 7: Sacrifice Rate of GUARD upon unlearning Phi-1.5B on 5% of all datapoints in TOFU. When unlearning a larger portion (compared to the portion of 1% in Table 6) of the dataset, GUARD generally induces a larger reduction on the sacrifice rate across all evaluation sets (Retain Set, Real Author Set, and Real World Set) and methods (GA, GD, KM, and PO). The largest improvement is observed on the Retain Set, where GUARD lowers $\rho_r$ by 23% over KM, $\rho_p$ by 26% over GA, and $\rho_t$ by 18.02% over GA. Notably, in the Real World Set, PO+*GUARD* leads to negative sacrifice rates across all evaluation metrics, suggesting that GUARD effectively mitigates unintended knowledge degradation and even improves model generalization.

| Methods | Retain Set | | | Real Author Set | | | Real World Set | | |
|---|---|---|---|---|---|---|---|---|---|
| | $\rho_r$ | $\rho_p$ | $\rho_t$ | $\rho_r$ | $\rho_p$ | $\rho_t$ | $\rho_r$ | $\rho_p$ | $\rho_t$ |
| GA | 97.96 | 92.47 | 101.97 | 40.90 | 19.97 | 20.52 | 74.41 | 20.53 | 21.10 |
| GA+GUARD | **79.68** | **73.74** | **83.95** | **27.73** | **13.22** | **15.16** | **53.10** | **14.05** | **16.10** |
| GD | 60.45 | 40.92 | 67.58 | 33.37 | 11.70 | 19.55 | 25.70 | 3.28 | 5.48 |
| GD+GUARD | **55.43** | **35.30** | **63.50** | **30.32** | **4.82** | **8.79** | **18.78** | **-5.17** | **-9.44** |
| KM | 97.96 | 97.55 | 99.72 | 40.90 | 17.33 | 17.81 | 74.41 | 15.56 | 15.99 |
| KM+GUARD | **74.65** | **72.35** | **97.08** | **28.73** | **13.00** | **17.56** | **58.10** | **10.79** | **14.58** |
| PO | 43.41 | 74.85 | 173.78 | -8.85 | 8.58 | 20.06 | -8.32 | -3.01 | -7.02 |
| PO+GUARD | **35.84** | **57.50** | **161.89** | **-13.17** | **5.95** | **16.91** | **-16.08** | **-17.42** | **-16.76** |

**Dataset-Specific Observations.** Across all datasets, splits, and evaluation metrics, GUARD achieves the most significant improvements on the Retain Set, which aligns with its design: the proxy data attribution is computed based on the alignment between the forget data and the overall retain data, directly benefiting retention performance. In contrast, improvements on the generalization sets (Real World Set and Real Author Set) are indirect. Since unlearning the forget set inherently reduces overfitting, we observe negative sacrifice rates in the Real World Set and Real Author Set, which suggest that GUARD not only prevents unintended knowledge degradation but may also enhance model generalization.

F.4   RESULTS FOR ICLM-7B ON THE MUSE BOOKS DATASET

We present the results of ICLM-7B on MUSE BOOKS dataset in Table 9.

F.5   COMPUTATIONAL EFFICIENCY ANALYSIS

To evaluate the computational overhead introduced by GUARD, we conducted runtime comparisons between standard Gradient Ascent (GA) and GA augmented with GUARD across different datasets

Table 8: Sacrifice Rate of GUARD upon unlearning Phi-1.5B on 10% of all datapoints in TOFU. When unlearning a larger portion of the dataset, GUARD generally induces an even larger reduction on the sacrifice rate across all evaluation sets (Retain Set, Real Author Set, and Real World Set) and methods (GA, GD, KM, and PO). The largest improvement is observed on the Retain Set, where GUARD lowers $\rho_r$ by 23.7% over PO, $\rho_p$ by 50.44% over PO, and $\rho_t$ by 54.75% over GA.

| Methods | Retain Set | | | Real Author Set | | | Real World Set | | |
|---|---|---|---|---|---|---|---|---|---|
| | $\rho_r$ | $\rho_p$ | $\rho_t$ | $\rho_r$ | $\rho_p$ | $\rho_t$ | $\rho_r$ | $\rho_p$ | $\rho_t$ |
| GA | 98.76 | 94.45 | 258.34 | 41.31 | 17.55 | 60.23 | 75.15 | 27.16 | 160.42 |
| GA+GUARD | **80.88** | **86.43** | **203.59** | **21.73** | **13.72** | **49.99** | **44.10** | **14.44** | **106.51** |
| GD | 73.97 | 97.90 | 183.63 | 39.34 | 11.16 | 13.26 | 57.21 | -5.54 | 9.39 |
| GD+GUARD | **69.32** | **82.29** | **168.55** | **29.05** | **10.03** | **3.60** | **53.54** | **-19.68** | **-17.97** |
| KM | 98.05 | 95.24 | 250.46 | 40.94 | 16.92 | 60.97 | 74.48 | 15.19 | 117.87 |
| KM+GUARD | **75.14** | **79.84** | **197.48** | **17.75** | **10.93** | **26.22** | **72.04** | **11.28** | **92.99** |
| PO | 76.84 | 78.48 | 245.73 | -6.14 | 9.80 | 22.31 | 4.53 | -1.32 | 21.38 |
| PO+GUARD | **53.19** | **28.04** | **224.45** | **-9.96** | **8.44** | **3.06** | **-2.71** | **-3.41** | **8.57** |

Table 9: Unlearning experiments with ICLM-7B on the MUSE BOOKS dataset.

| Methods | Forget Set VerbMem ($\downarrow$) | Forget Set KnowMem ($\downarrow$) | PrivLeak ($\in [-5\%, 5\%]$) | Retain Set KnowMem ($\uparrow$) |
|---|---|---|---|---|
| | | BOOKS | | |
| w/o Unlearn | 99.8 | 59.4 | $-57.5$ | 66.9 |
| GA | 0.0 | 0.0 | $-$**25.0** | 0.0 |
| GA | **0.0** | **0.0** | $-31.3$ | **10.3** |
| KM | **15.4** | 22.5 | $-$**39.1** | 35.9 |
| KM + GUARD | 16.0 | **20.3** | $-42.2$ | **47.2** |
| GD | 0.0 | 0.0 | $-29.1$ | 11.6 |
| GD + GUARD | **0.0** | **0.0** | $-26.6$ | **28.1** |
| NPO | 0.0 | 0.0 | -32.3 | 24.5 |
| SimNPO | 0.0 | 0.0 | **-22.6** | 47.8 |
| FLAT (TV) | 0.0 | 0.0 | -45.2 | 28.9 |
| SimImP | 0.0 | 0.0 | -26.4 | 22.5 |
| NPO + GUARD | **0.0** | **0.0** | -38.16 | **52.71** |

and model architectures. Table 10 presents the runtime measurements for single epoch unlearning experiments conducted on our experimental setup with 4 NVIDIA Tesla V100-SXM2 GPUs.

The runtime comparison between GA and GA + GUARD provides a representative assessment of GUARD's computational overhead. Importantly, the additional time introduced by GUARD primarily stems from the data attribution computation phase, which is independent of the choice of baseline unlearning method. This means that the overhead observed with GA would be similarly applicable when GUARD is integrated with other baseline methods.

A critical aspect of GUARD's computational efficiency lies in the nature of its data attribution computation. The attribution score calculation described in Eq. 2 serves as a preprocessing step that is performed only once for each model parameters-dataset combination. Crucially, this computation is independent of the number of unlearning epochs and the frequency of unlearning operations. Once the attribution scores are computed for a given forget set, they can be reused across multiple unlearning sessions or when experimenting with different hyperparameters, amortizing the computational cost over multiple uses.

In conclusion, GUARD demonstrates practical computational efficiency with minimal time overhead. The one-time nature of the attribution computation, combined with the modest runtime increases observed across diverse experimental settings, establishes GUARD as a computationally viable enhancement to existing unlearning methods without imposing prohibitive computational costs.

Table 10: Runtime of GUARD Unlearning for Single Epoch

| Dataset | Model | Forget Proportion/Set | Runtime of GA (min) | Runtime of GA + GUARD (min) |
|---------|-------|----------------------|---------------------|------------------------------|
| TOFU | LLaMA-2-7B | 1% | 4.2 | 5.1 |
| | | 5% | 14.4 | 16.6 |
| | | 10% | 32.3 | 35.5 |
| | Phi-1.5B | 1% | 2.8 | 3.0 |
| | | 5% | 6.1 | 6.8 |
| | | 10% | 14.5 | 15.6 |
| MUSE | LLaMA-2-7B | NEWS | 83.7 | 91.2 |
| | ICLM-7B | BOOKS | 116.0 | 127.9 |

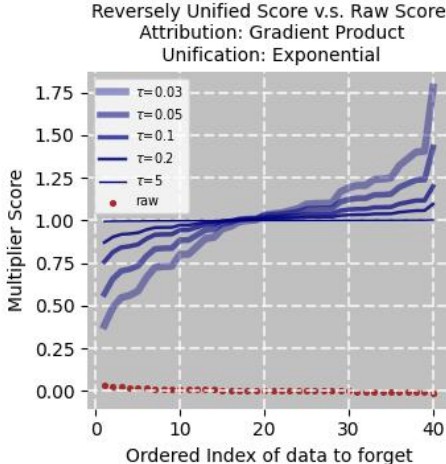

Figure 3: Raw Score and reversely unified score given by different attribution and unification methods.

### F.6 PROXY FOR DATA ATTRIBUTION

For the unification methods, the reversely unified scores controlled by temperature are plotted in the right two plots of Fig. 3. Exponential unification provides a smoother attribution distribution at a lower temperature.

### F.7 HYPERPARAMETER SENSITIVITY

We present the unlearning effect of GUARD in Fig. 4, where we use the example of unlearning Phi-1.5B on TOFU (Forget Split = 1%). In Fig. 4, the x-axis use $-\log(\tau)$ to depict various $\tau$ *for unification*. Fig. 4 shows the performance across three evaluation metrics: ROUGE-1, ROUGE-1 drop $\Delta_{\text{ROUGE-1}}$, and Sacrifice Rate $\rho^{ul}$. The results demonstrate that our method significantly outperforms the baseline in maintaining the desired knowledge.

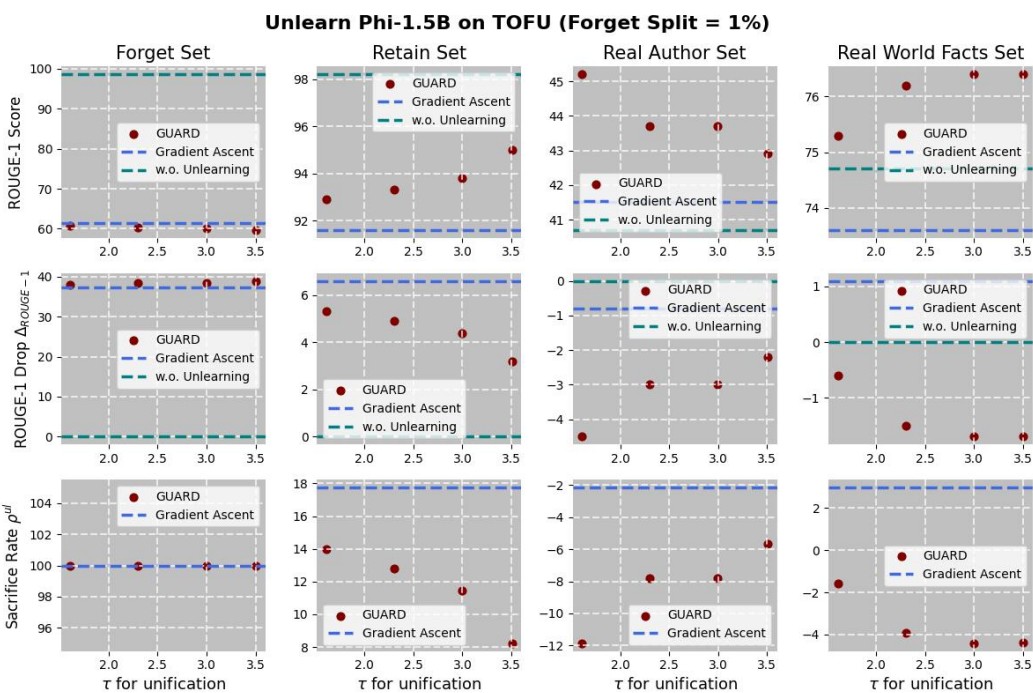

Figure 4: Hyperparameter Sensitivity Analysis.

