# OpenReview forum: "GUARD: Guided Unlearning and Retention via Data Attribution for Large Language Models"
_ICLR.cc/2026/Conference — ICLR 2026 Conference Desk Rejected Submission_

### Official Review · Reviewer_1Uua · 2025-10-31

**Soundness:** 3
**Presentation:** 2
**Contribution:** 2
**Rating:** 4
**Confidence:** 4

**Summary:**

This paper addresses a critical challenge in machine unlearning for large language models (LLMs)—the trade-off between effective forgetting of targeted data and retention of useful knowledge. The proposed method, GUARD (Guided Unlearning and Retention via Data Attribution), introduces a data attribution–based framework that adaptively assigns unlearning weights inversely proportional to each sample’s influence on the retained data. Results demonstrate substantial improvements in knowledge retention while maintaining similar forgetting performance.

**Strengths:**

1. The attribution-driven unlearning method is underexplored in the literature and the proposed method is simple.

2. The experiments show that the method is effective compared to baselines.

3. The authors provide proofs supporting improvements in retention efficiency.

**Weaknesses:**

1. The writing could be improved. For example the introduction is lengthy and reduant to some extent, which reduces clarity and readability.

2. No other data attribution methods are tested. It would strengthen the paper to explore different attribution measures or analyze sensitivity to gradient noise.

**Questions:**

See weaknesses.

Since the proposed method uses the inner product of gradients as the proxy, I'm wondering why not just use the similarity of embeddings rather than the gradients? What would be the results compared to the proposed method?

---

> ### Author Response · Authors · 2025-11-20
>
> For weakness 1:
>
> We sincerely appreciate the reviewer's constructive feedback on presentation quality. We acknowledge that the introduction could be more concise and structured to enhance clarity and readability. In the revision, we will streamline Section 1 by removing redundant content, reorganizing the motivation to flow more directly from problem statement to solution, and sharpening the exposition of our key contributions. We will also review the entire manuscript for similar opportunities to improve clarity and reduce unnecessary length while preserving technical rigor. Nevertheless, this being said, we would also like to remark that some redundancy can be helpful in order to “get the message across” and remind the reviewers about what has been explained before (which was the reason we did include some repeated comments). Please note that this small redundancy was not used for page padding, as this is evidently not the case - our Supplement is comprehensive and many of the results therein could have been moved to the main text.
>
> For weakness 2 and question:
>
> While data attribution is well-established in general machine learning, methods with acceptable computational complexity for LLM unlearning are remarkably scarce - classical approaches like influence functions require Hessian inversion, TracIn requires storing all checkpoints, while TRAK demands ensemble approximations, all prohibitively expensive for billion-parameter models. To address this concern, we conducted experiments with embedding similarity-based attribution, another first-order method  that is tractable for LLMs. The key distinction is that embeddings capture semantic similarity in representation space, while gradients capture directional influence on the model parameters: for unlearning, what matters is not whether the samples are semantically similar, but whether removing one will degrade performance on the other. Our additional results(the results are detailed in our response to Reviewer u1aC) generated to investigate the issue raised by the reviewer validate this intuition: NPO + GUARD (gradient-based, ρ_r=61.12, Truth Ratio=71.94) substantially outperforms NPO + Embedding Similarity (ρ_r=80.22, Truth Ratio=56.77), demonstrating that gradient alignment ⟨g_r, g_i⟩ better captures retention-relevant influence. We will incorporate this new comparison and complexity analysis into the revised manuscript.

---

### Official Review · Reviewer_u1aC · 2025-11-01

**Soundness:** 3
**Presentation:** 3
**Contribution:** 3
**Rating:** 4
**Confidence:** 3

**Summary:**

GUARD is a retention-aware unlearning method for LLMs. It scores each forget sample by the inner product between its gradient and the average retain gradient, then applies temperature-controlled inverse weights to focus unlearning where it least harms retention. Theory shows reduced retain loss, comparable forgetting, and a better retention–forget tradeoff. Experiments on LLaMA 2 7B across several objectives consistently improve retain performance with similar or better forgetting, and the approach is simple and efficient to plug into existing pipelines.

**Strengths:**

1. Recasts unlearning as retention aware weighting using a gradient alignment score, directly aligned with the goal of preserving retained knowledge.
2. GUARD pipeline shows clear properties and theory that guarantees lower retain loss, comparable forgetting, and a better retention–forget tradeoff.
3. Proposed GUARD pipeline gains across strong baselines and metrics, with large improvements on Retain while keeping Forget performance on target, also quite easy to plug into existing procedures.
4. The description of the whole paper is quite clear.

**Weaknesses:**

1. The core score is the inner product between a forget sample gradient and the average retain gradient. So I am curious about the effect of the inverse weighting by deriving it from an explicit objective, for example, maximizing loss on the forget set subject to a first-order constraint on the retain set, and present the solution through a Lagrangian or projection analysis.
2. In the experiment section, the comparison is mainly focused on the current MU methods and with GUARD pipeline, and since the authors mentioned the novelty of GUARD pipeline with the current data attribution methods, why not add the comparison between these methods and GUARD in the experiment section.
3. The experiment misses the sensitivity ablation of some parameters of GUARD training. In addition, I am curious about whether introducing GUARD shifts the optimal settings of the underlying unlearning methods. For example, does the preferred NPO $\beta$ change with GUARD, and how do key parameters for other unlearning algorithms behave before vs. after applying GUARD?
4. Prior work suggests RMU is a more stable unlearning algorithm than NPO. Including RMU as an additional baseline would strengthen the empirical validity of the proposed method.

**Questions:**

Please see the weakness section, and I will consider raising the score if the authors address the problems clearly

---

> ### Author Response · Authors · 2025-11-20
>
> For weakness 1:
>
> This is an insightful suggestion that we appreciate. The inverse weighting in GUARD can indeed be rigorously motivated through a first-order constrained optimization framework. Consider the linearized unlearning problem: max_Δ g_f^⊤ Δ subject to g_r^⊤ Δ ≤ ε and ||Δ|| ≤ R, where g_f and g_r are the averaged gradients over forget and retain sets, ε bounds the retain loss increase, and R enforces a trust region. Forming the Lagrangian and solving for the optimal update yields Δ* = (1/η)(g_f - λg_r), where the Lagrangian multiplier λ = (g_r^⊤ g_f - ηε)/(g_r^⊤ g_r) when the retain constraint is active. Critically, the inner product g_r^⊤ g_f appearing in this closed-form solution is precisely our attribution score—it quantifies how much the forget gradient conflicts with the retention constraint. Geometrically, this corresponds to projecting g_f onto the subspace orthogonal to g_r when ε = 0. GUARD extends this global constraint to adaptive per-sample penalties: samples with high ⟨g_r, g_i⟩ receive lower unlearning weights, effectively enforcing heterogeneous retention constraints across the forget set. The temperature τ in Equation (3) modulates this adjustment strength analogously to the Lagrangian multiplier. While we focused on the intuitive gradient-alignment interpretation in the current manuscript, we acknowledge that this optimization-theoretic perspective strengthens the theoretical foundation and will incorporate this derivation in the revision, potentially connecting it to projection-based unlearning frameworks.
>
> For weakness 2:
>
> We thank the reviewer for raising an excellent point about comparing with existing attribution methods. However, it is crucial to recognize that while data attribution is a mature field in general machine learning, methods with acceptable computational complexity for LLM unlearning are remarkably scarce. Classical approaches like influence functions (Koh & Liang, 2017) require Hessian inversion (O(d³) for d parameters), TracIn (Pruthi et al., 2020) requires storing all training checkpoints, and TRAK (Park et al., 2023) demands ensemble approximations, all of which are prohibitively expensive for billion-parameter models. GUARD was specifically designed to address this complexity bottleneck through a first-order gradient approximation (O(d) per sample). To validate our design choice, we have conducted additional experiments comparing GUARD with embedding similarity-based attribution, another first-order method that measures sample influence via representation space distances. As shown in Table 1 and Table 2, NPO + GUARD (ρ_r=61.12, Truth Ratio=71.94) substantially outperforms NPO + Embedding Similarity (ρ_r=80.22, Truth Ratio=56.77), demonstrating that gradient-based alignment better captures retention-relevant influence than embedding proximity. We will include this comparison and a more in-depth discussion of computational tractability in the revised manuscript.
>
> | Methods                   | **Retain Set ↓** |          |          | **Real Author Set ↓** |          |          | **Real World Set ↓** |          |          |
> |---------------------------|------------------|----------|----------|------------------------|----------|----------|------------------------|----------|----------|
> |                           | ρr              | ρp      | ρt      | ρr                    | ρp      | ρt      | ρr                    | ρp      | ρt      |
> | NPO + GUARD               | 61.12            | 78.39    | 118.47   | 23.88                  | 4.24     | 11.62    | -7.87                  | -4.55    | -13.69   |
> | NPO + Embed. Similarity   | 80.22            | 97.28    | 188.43   | 39.86                  | 41.33    | 88.74    | 6.25                   | 30.17    | 15.40    |
> | RMU                       | 70.25            | 81.66    | 135.79   | 30.01                  | 12.28    | 56.96    | 6.73                   | 7.30     | -4.69    |
>
> Table 1 Results of Sacrifice Rate with LLaMA-2-7B unlearning on 10% of all data points in TOFU
>
> | Methods                      | **Retain Set ↑** |          |          | **Forget Set ↓** |          |          |
> |------------------------------|------------------|----------|----------|-------------------|----------|----------|
> |                              | ROUGE-L ↑        | Prob. ↑  | T. Ratio ↑ | ROUGE-L ↓        | Prob. ↓  | T. Ratio ↓ |
> | NPO + GUARD                  | 86.25            | 60.54    | 71.94    | 31.79             | 42.57    | 63.26     |
> | NPO + Embed. Similarity   | 72.52            | 52.01    | 56.77    | 32.96             | 43.51    | 63.68     |
> | RMU                          | 78.20            | 56.77    | 65.24    | 31.12             | 42.01    | 62.09     |
>
> Table 2 Results of Absolute Performance with LLaMA-2-7B unlearning on 10% of all data points in TOFU

---

> ### Author Response · Authors · 2025-11-20
>
> For weakness 3:
>
> We thank the reviewer for this thoughtful observation. To clarify: when applying GUARD to NPO (or any baseline method), we intentionally keep the baseline's hyperparameters (learning rate, β, etc.) identical to their standalone optimal settings - we only introduce GUARD's temperature τ as an additional hyperparameter. This design choice ensures that GUARD acts as a modular enhancement rather than requiring method-specific retuning. Nevertheless, the reviewer's intuition is correct: the hyperparameters optimized for standalone NPO may not be optimal when combined with GUARD's attribution-driven weighting, suggesting that our current results represent a lower bound on the potential performance of GUARD. Jointly optimizing both NPO's β and GUARD's τ would likely  result in further performance improvements, but we prioritize demonstrating that GUARD provides gains even without such co-optimization. Appendix F.7 presents the sensitivity analysis for τ across multiple baselines, showing consistent improvements across a wide range of values, which confirms GUARD's robustness. We will explain the reasons behind this conservative experimental design more clearly in the revision.
>
> For weakness 4:
>
> We appreciate the reviewer's suggestion to include RMU, which indeed represents a complementary approach emphasizing robustness. Following this recommendation, we have conducted additional experiments with RMU as shown in Table 1 and Table 2. The results demonstrate that while RMU (ρ_r=70.25, ρ_p=81.66, ρ_t=135.79) provides reasonable unlearning performance, NPO + GUARD (ρ_r=61.12, ρ_p=78.39, ρ_t=118.47) achieves superior retention across all metrics on the Retain Set, with particularly notable improvements in the Truth Ratio (71.94 vs. 65.24). This comparison strengthens the empirical validity of our approach and confirms that GUARD's attribution-driven strategy offers advantages beyond what architectural or algorithmic innovations alone could provide. We will incorporate the RMU results into the main experimental comparisons in the revised manuscript. We once again thank the reviewer for this valuable suggestion.

---

### Official Review · Reviewer_sr5u · 2025-11-01

**Soundness:** 2
**Presentation:** 2
**Contribution:** 2
**Rating:** 2
**Confidence:** 3

**Summary:**

This paper investigates LLM unlearning from a data attribution perspective. It proposes a data attribution algorithm GUARD, which first uses gradient information to estimate the influence of each data sample in the retain/forget data. Then performs unlearning training using the gradient-information weighted loss on each sample. The paper presents a theoretical guarantee to prove its effectiveness, and shows better performance than other baselines in the expeirments.

**Strengths:**

* The proposed weighting method can be applied to many different LLM unlearning loss, and it seems to bring benefit to multiple method in the experiment.
* The paper presents a theoratical gurantee about the weighting's effectiveness for unlearning training.

**Weaknesses:**

* Potential unreasonable unlearning setting. Equation (2) computation on the weighting depends on original model $\theta_0$ gradient on the $D_r$ dataset and forget set example. This does not look reasonable to me, since typical unlearning does not assume access to the pre-trained model weight before fine-tuning on the knowledge data.
* Potential unreasonable data assumption. Assumption 1 (line 291) mentions condition 1, with $<\bar{g_r}, \bar{g_f}>0$, which seems abrupt and lacks sufficient motivation why is this case. Authors should provide more justification on why this holds.
* Potential robustness concern on retain data selection. Since the attribution relies on both the forget data (user request) and retain data (may be arbitrary), this paper lacks discussion on the robustness of retain data selection, and how the retain data should be constructed.
* Non-standard unlearning hyper-parameter selection. Appendix Page 26 presents the forgetting training hyper-parameter only involving 1 epoch, and batch size 1, which is not the common practice in previous work, that mostly involving more epoch training and best checkpoint selection like those in [1, 2].

[1] Fan, Chongyu, et al. "Simplicity prevails: Rethinking negative preference optimization for llm unlearning." arXiv preprint arXiv:2410.07163 (2024).

[2] Maini, Pratyush, et al. "Tofu: A task of fictitious unlearning for llms." arXiv preprint arXiv:2401.06121 (2024).

**Questions:**

* Appendix computation cost compares normal training and GUARD-guided weighting unlearning training, however, the main computation cost in obtaining the gradient information on all data sample.

---

> ### Author Response · Authors · 2025-11-20
>
> For weakness 1:
>
> We respectfully disagree with the reviewer. To clarify, it is possible that the reviewer has misunderstood the order of execution steps of our method. The standard LLM unlearning pipeline proceeds through the following processing steps: (1) Start with a pre-trained model → (2) Fine-tune on knowledge data D₀ = D_f ∪ D_r, which gives rise to model θ₀ → (3) Compute data attribution on θ₀ → (4) Perform unlearning on D_f. Our attribution computation (Equation 2) occurs at step (3), after fine-tuning but before unlearning, when organizations naturally have full access to both the fine-tuned checkpoint θ₀ and the training data. Using TOFU as an example: we fine-tuned pre-trained LLaMA-2 on the TOFU dataset to obtain θ₀, then compute the attribution scores once, then apply unlearning on the forget set — note again that this is the *standard protocol* followed by all unlearning benchmarking methods. There is no requirement to access pre-trained weights, and the reviewer's concern is unjustified.
>
> For weakness 2:
>
> We thank the reviewer for asking why we assumed that ⟨g_r, g_f⟩ > 0. This condition reflects the fundamental reality of knowledge entanglement in LLM training: when a model is fine-tuned on a coherent corpus, samples naturally share semantic features, linguistic patterns, or topical overlap, leading to positively correlated gradients. If this condition did not hold (⟨g_r, g_f⟩ ≈ 0), the forget and retain sets would already be orthogonal in the gradient space, making unlearning a straightforward task: simply applying gradient ascent would not harm retention, and no specialized methods would be needed. The extensive empirical evidence of utility degradation in prior unlearning work [1] confirms that ⟨g_r, g_f⟩ > 0 is the norm rather than the exception. Our assumption formalizes the very problem setting that motivates retention-aware unlearning methods.
>
> [1] Zhang et al., "Catastrophic failure of llm unlearning via quantization." ICLR 2025.
>
> For weakness 3:
>
> We thank the reviewer for raising their concern about the retain set robustness/construction. It is worth emphasizing that GUARD operates as a general framework applicable to any user-specified retain sets, whether it be the full complement of training data minus the forget set, a curated subset deemed critical, or surrogate corpora. To empirically validate this robustness, we conducted experiments comparing the use of randomly selected 20% versus 100% of the retain set for evaluating the data attribution method—the results are detailed in our response to Reviewer 9Kbs, demonstrating that GUARD maintains substantial retention improvements even with limited retain data. The robustness issues that the reviewer raises do not apply to GUARD — our method adaptively weights samples based on whatever retain set is provided, making it robust to different retention priorities. Prior work on data selection (Fisher information-based sampling, coreset methods [2, 3]) can be seamlessly integrated to construct representative retain sets. We acknowledge that this discussion would clarify the properties of the GUARD method and strengthen the paper and we will definitely include the reviewer's question/our response regarding the retain set construction strategies in the revision. Also, we must note that standardized benchmarks we used for testing, such as TOFU and MUSE, have predefined canonical retain sets for reproducibility.
>
> [2] Liu et al., “Fisher Information-based Efficient Curriculum Federated Learning with Large Language Models.” EMNLP 2024.
>
> [3] Mirzasoleiman et al., “Coresets for Data-efficient Training of Machine Learning Models.” ICML 2020.
>
> For weakness 4:
>
> We appreciate the reviewer's reference to best practices in the 2 papers. Our hyperparameter choices reflect two practical constraints: (1) The setting of batch size 1 represents the most rigorous / challenging test case for maintaining model robustness during unlearning: a smaller batch size leads to a stronger, localized overfitting on forget set unlearning, making it substantially more difficult for LLMs to maintain strong performance on the general data. A successful mitigation in this most challenging scenario validates the method’s effectiveness across general cases. (2) A single epoch is adopted because extending beyond one epoch causes catastrophic utility collapse in nearly all baseline methods — the models forget not only the target data but also essential linguistic capabilities, rendering them unusable in practice. This is consistent with the TOFU benchmark protocol, which *explicitly requires the use of single-epoch* unlearning for this reason. Our goal is controlled comparison under realistic resource constraints, not exhaustive hyperparameter optimization. Critically, identical settings are applied to both baselines and GUARD variants, ensuring that performance differences arise solely from our attribution-driven weighting mechanism.

---

> > ### Comment · Reviewer_sr5u · 2025-11-21
> >
> > Thank authors for the clarification. However, I am still confused about the batch size 1, and one-epoch training.
> >
> > For BS=1 choice, I don't agree that this resembles one-by-one user request since for each data sample, the optimization is not fully done, meaning that it does not converge in the current training setting. Instead, it's similar to perform one-step gradient update for each data sample.
> >
> > For one-epoch training, I think prior works in addition reports a utility-unlearn performance trade-off trajectory through multi-epoch training to help understand the stability of different training algorithm. Simply stops the training at the end of first epoch seems to be abrupt design choice to me since based on my personal experience with ToFU data, some of the best checkpoints can be at the end of epoch 2 or 3 instead of epoch 1.

---

> > > ### Author Response · Authors · 2025-11-27
> > >
> > > We thank the reviewer for the response. To clarify, we provide a detailed explanation of why our specific configuration (BS=1, Epoch=1) was chosen to rigorously evaluate robustness and practical utility, and how we determined the optimal stopping point based on a specific utility threshold.
> > > 1. On One-Epoch Training (Observed Utility-Forgetting Trajectory) Regarding the choice of 1 epoch versus multi-epoch training, our decision was directly informed by the utility degradation trajectory and a strict practical utility threshold:
> > >
> > > Observation of Rapid Degradation: During our preliminary hyperparameter tuning phase, we monitored the model's performance beyond the first epoch. We consistently observed that under the standard fine-tuning learning rates used for unlearning, extending training to Epoch 2 or 3 led to a disproportionate collapse in Retain Set utility, rendering the model practically unusable.
> > >
> > > Utility Threshold and Practical Value: In practical applications, we define a model as lacking practical value if its Truth Ratio on the Retain Set falls below 50% (i.e., the model fails to answer even half of the retained knowledge questions correctly). Consequently, we utilized 50% as our critical selection threshold: specifically, we selected the epoch that exhibited the best trade-off between forgetting and retention, provided that the Retain Set Truth Ratio remained above 50%. if the metric drops below this level at the first epoch, we stop training to prevent further destruction of knowledge.
> > >
> > > Evidence from Table 2: As reported in Table 2 of our manuscript, standard baselines such as GA (19.32%), KM (26.21%), and PO (39.83%) already drop below this 50% Truth Ratio threshold after just one epoch. This confirms that extending training to a second epoch would only cause further catastrophic collapse. To ensure a fair and controlled comparison, we standardized the stopping point at Epoch=1 for all methods to evaluate performance at this critical limit of usability.
> > >
> > > 2. On Batch Size = 1 (Robustness Stress-Test) We agree with the reviewer’s precise observation that BS=1 represents a stochastic, high-variance update step rather than a converged optimization per sample. However, this was a deliberate design choice to stress-test the robustness of the unlearning methods:
> > >
> > > High-Noise Regime: In real-world scenarios, unlearning often involves noisy or sparse signals. BS=1 creates a challenging optimization landscape where individual gradient updates can be orthogonal or conflicting with the Retain Set manifold.
> > >
> > > Validating GUARD: This setting perfectly isolates the contribution of GUARD. By explicitly reweighting these noisy, high-variance updates based on data attribution, GUARD prevents the "bad steps" from damaging the Retain Set. The fact that GUARD significantly outperforms baselines in this high-noise regime (BS=1) strongly evidences the effectiveness of our attribution-driven objective.
> > >
> > > We thank you again for your valuable feedback. We realize that we did not explicitly detail the rationale for the epoch and batch size selection (specifically the 50% utility threshold) in the main text. We will incorporate this detailed justification into the experimental setup section of the revised paper.

---

> ### Author Response · Authors · 2025-11-20
>
> For the question:
>
> We thank the reviewer for the question. We would like to clarify that the computation time reported for GUARD-guided unlearning has already included the cost of calculating the necessary gradient information. Therefore, the overhead presented is a true reflection of the total integrated cost, including the required gradient computations. As detailed in Appendix F.5 and Table 10, GUARD introduces minimal overhead (15-20% runtime increase) because the attribution computation is a one-time preprocessing step that amortizes across multiple unlearning operations. Once computed for a given dataset, these scores can be reused for different forget subsets, hyperparameter experiments, or repeated unlearning requests without recalculation. The per-sample gradient computation (Equation 2) requires only forward-backward passes without Hessian estimation or model retraining, making it far more efficient than conventional influence function methods. For context, on TOFU 10% with LLaMA-2-7B, GUARD adds merely 3.2 minutes to a 32.3-minute baseline—a negligible cost for the substantial retention improvements we demonstrated.

---

### Official Review · Reviewer_9Kbs · 2025-11-01

**Soundness:** 3
**Presentation:** 3
**Contribution:** 3
**Rating:** 6
**Confidence:** 3

**Summary:**

This paper proposes GUARD, a method for LLM unlearning based on data attribution scores. Specifically, instead of using uniform weights on all data samples, the authors calculate the attribution score for a particular forget sample as the inner product of its gradient and the average gradient over the retain set. Then the weight for each forget sample is assigned to be inversely proportional to its attribution score, so that samples that might lead to a large performance drop on the retain set are assigned smaller weights. The authors further present theoretical results showing that GUARD can improve the sacrifice rate over the baseline gradient ascent method. Experiments on two datasets show that GUARD outperforms strong baselines.

**Strengths:**

1. The proposed method is simple intuitively, and experiments show that it outperforms strong baselines, especially on the retain performance.
2. The results show that using the proposed sample weights consistently outperforms the uniform weights when applied to different unlearning methods, models, and datasets.
3. The proposed method is theoretically grounded.

**Weaknesses:**

1. The method assumes access to the retain set in order to calculate the attribution scores. However, what if we don't know what questions are in the retain set? Does the method still work if we use a general corpus, such as a subset of the pre-training corpus, to calculate the attribution score? Are there any ways to obtain some surrogates for the retain set?
2. The method needs to first calculate gradients over the retain and forget set, which is more expensive than baselines. How much additional cost does this bring?

**Questions:**

Please see weaknesses.

---

> ### Author Response · Authors · 2025-11-20
>
> For weakness 1:
>
> This is an excellent question. The reviewer is correctly raising the issue of retain set maintenance and knowledge, since many LLMs never store their retain sets, either due to their excessively large volumes, privacy concerns or both. Nevertheless, it is worth pointing out the following:
> Smaller-scale, privately maintained LLM models, for which this work is mostly tailored to, do indeed store their retain set. As the reviewer knows, biomedical and biological databases and corresponding LLMs are required to retain patient data for roughly 20 years. The same is the case for financial and other institutions. A quick check reveals that, for example, BloombergGPT is actively storing its training datasets, and so do some biomedical institutions that maintain their own biomedical LLMs. Nevertheless, even when the full retain set is unavailable, the distribution of gradients is what matters and can be estimated using easily available public proxies (surrogates). Since the retain set fundamentally serves to reflect domain knowledge and preserve model capability, viable surrogates include pre-training subsets, public domain-aligned datasets, or synthetic data representative of the target domain. Prior literature encompasses many examples of using such surrogate retain sets, and methods like Fisher-sensitive sampling can identify smaller-scale subsets representative of the true retain gradients. To empirically validate this robustness, we conducted experiments using both 100% and 20% of the retain set to evaluate the data attribution method—as shown in Table 1 and Table 2, even with only 20% of the retain set, GUARD still achieves substantial retention improvements compared to baselines. Hence, our answer is that there is no practical challenge (except for additional computational investments) to practically deploy the method even if the full retain set is not known. We will be happy to elaborate on these surrogate methods in the revised manuscript, but would not like this to be the focal point of the paper since the current work already contains numerous new contributions.
>
> | Methods                         | Retain Set(↓) |          |          | Real Author Set(↓) |          |          | Real World Set(↓) |          |          |
> |---------------------------------|------------------|----------|----------|------------------------|----------|----------|------------------------|----------|----------|
> |                                 | ρr              | ρp      | ρt      | ρr                    | ρp      | ρt      | ρr                    | ρp      | ρt      |
> | NPO + GUARD (100% retain set)   | 61.12            | 78.39    | 118.47   | 23.88                  | 4.24     | 11.62    | -7.87                  | -4.55    | -13.69   |
> | NPO + GUARD (20% retain set)    | 64.62            | 80.82    | 125.30   | 24.02                  | 4.35     | 13.09    | -6.99                  | -3.76    | -11.85   |
> | NPO                             | 85.58            | 106.24   | 201.33   | 42.91                  | 42.47    | 101.43   | 8.11                   | 33.24    | 20.32    |
>
> Table 1 Results of Sacrifice Rate with LLaMA-2-7B unlearning on 10% of all data points in TOFU
>
>
> | Methods                       | Retain Set (↑)                  |                         |                         | Forget Set (↓)                  |                         |                         |
> |-------------------------------|----------------------------------|--------------------------|--------------------------|----------------------------------|--------------------------|--------------------------|
> |                               | ROUGE-L ↑                       | Prob. ↑                 | T. Ratio ↑               | ROUGE-L ↓                       | Prob. ↓                 | T. Ratio ↓               |
> | NPO + GUARD (100% retain set) | 86.25                            | 60.54                   | 71.94                   | 31.79                            | 42.57                   | 63.26                   |
> | NPO + GUARD (20% retain set)  | 84.08                            | 58.77                   | 69.15                   | 31.52                            | 43.20                   | 63.88                   |
> | NPO                           | 70.01                            | 48.30                   | 55.26                   | 32.12                            | 42.71                   | 63.89                   |
>
> Table 2 Results of Absolute Performance with LLaMA-2-7B unlearning on 10% of all data points in TOFU

---

> ### Author Response · Authors · 2025-11-20
>
> For weakness 2:
>
> This is another important question which we already addressed in the paper, but the reviewer may have missed it since it appears in the Supplement. Please see Table 13 for runtime comparisons of GA vs. GA-GUARD (i.e., GA combined with data attribution). You can see that the increase in runtime is roughly 15-20%. Besides, we would like to clarify that in realistic settings, unlearning requests are typically proposed by users on a small dataset, therefore such computation budget increase is generally very small when weighed against the significant and provable security benefits.

---

### Note · Program_Chairs · 2026-01-17
**Submission Desk Rejected by Program Chairs**

The following references in this submission do not refer to real documents and/or have major errors in bibliographic information:

 Wei Chen and Zhi Yang. Machine unlearning in the fine-tuned models of large language models. AI & Machine Learning, 12(6):34-48, 202
Rishabh Kumar, Sanjay Gupta, and Pooja Sharma. Machine unlearning for large language models: Challenges and opportunities. Journal of Machine Learning Research, 23(12):1-25, 2022.
Karan Maini and et al. Gradient ascent unlearning in large language models. NeurIPS, 2024.